# CRISPR-mediated gene silencing reveals involvement of the archaeal S-layer in cell division and virus infection

Isabelle Anna Zink[1,4], Kevin Pfeifer[1,2,4], Erika Wimmer[1], Uwe B. Sleytr[3], Bernhard Schuster[2] & Christa Schleper [1]*

The S-layer is a proteinaceous surface lattice found in the cell envelope of bacteria and archaea. In most archaea, a glycosylated S-layer constitutes the sole cell wall and there is evidence that it contributes to cell shape maintenance and stress resilience. Here we use a gene-knockdown technology based on an endogenous CRISPR type III complex to gradually silence *slaB*, which encodes the S-layer membrane anchor in the hyperthermophilic archaeon *Sulfolobus solfataricus*. Silenced cells exhibit a reduced or peeled-off S-layer lattice, cell shape alterations and decreased surface glycosylation. These cells barely propagate but increase in diameter and DNA content, indicating impaired cell division; their phenotypes can be rescued through genetic complementation. Furthermore, S-layer depleted cells are less susceptible to infection with the virus SSV1. Our study highlights the usefulness of the CRISPR type III system for gene silencing in archaea, and supports that an intact S-layer is important for cell division and virus susceptibility.

[1] Archaea Biology and Ecogenomics Division, Althanstraße 14, University of Vienna, A-1090 Vienna, Austria. [2] Institute for Synthetic Bioarchitectures, Muthgasse 11/II, University of Natural Resources and Life Sciences, A-1190 Vienna, Austria. [3] Institute of Biophysics, Muthgasse 11/II, University of Natural Resources and Life Sciences, A-1190 Vienna, Austria. [4] These authors contributed equally: Isabelle Anna Zink, Kevin Pfeifer. *email: christa.schleper@univie.ac.at

Monomolecular arrays of protein or glycoprotein subunits forming surface layers (termed S-layers[1]) represent an almost universal feature of archaeal envelopes and have been observed in species of nearly every taxonomical group of walled bacteria[2,3]. In contrast to the complex bacterial peptidoglycan or the pseudomurein found in some methanogenic archaea, the S-layer constitutes a rather simple cell wall, as it is composed of maximally two different protein species, which self-assemble to highly regular arrays of specific lattice patterns (i.e., p1–p6 symmetries), covering the whole cell[3]. In Gram+ and Gram− bacteria S-layer units bind to the peptidoglycan containing layer and lipopolysaccharides of the outer membrane, respectively, hence representing an outermost envelope layer of a complex cell wall structure[4]. This is a major difference to most archaea, where the S-layer serves as the sole cell wall component, directly binding the cytoplasmic membrane[2]. Binding is either achieved by a hydrophobic anchoring as in hyperthermophilic *Sulfolobus*, or via a covalent lipid–linkage, which seems predominant in halophilic archaea[5]. Forming dome-like protrusions, S-layer proteins create a pseudo-periplasmatic space[6], which was recently shown to stabilize vital cellular processes, such as ion flux[7]. Although some bacteria spontaneously lose their S-layer under laboratory conditions and even overgrow wild-type cells, spontaneous S-layer null-mutants have never been reported for any archaeon[8], indicating the importance of the structure to these organisms. All investigated archaeal S-layer proteins are heavily glycosylated and therefore many of the functions of this surface protein might be mediated by their glycans[9]. In particular, loss or alteration of the *N*-glycan structure caused growth retardation under elevated salt conditions and compromised protein secretion of archaeal glycosylation mutant cells in *Haloferax volcanii* and *Sulfolobus acidocaldarius*[10–13]. In some of these mutants, an aberrant or perturbed S-layer lattice was observed, which seemed more susceptible to proteolytic degradation, indicating a protective and stabilizing role of the glycosylation[10,12]. However, as surface proteins can share same *N*-glycan structures, pleiotropic effects of glycosylation breakdown on additional surface proteins cannot be excluded[10,11]. Only few studies reported about phenotypical changes of cells from which the whole S-layer glycoprotein was removed, which might represent the only approach to pin down its functional role in the cell. A primary function ascribed is maintenance of cell shape[14–16], as dramatic structural changes were observed in rod-shaped archaea[17–19], as well as the triangularly-shaped *Haloarcula japonica*[20] which all became small, spherical protoplasts after S-layer loss in ion-reduced media. Correspondingly, morphological changes were observed in *Haloferax volcanii* cells defective in the peptidase ArtA, which failed to remove the transmembrane domain of the S-layer glycoprotein, leading to an instable assembly[21,22]. In *Sulfolobus islandicus*, deletion of the S-layer genes resulted in stripped or partially stripped cells, and alterations in cell morphology, cell division, and sensitivity to hyperosmotic stress[23,24]. In line with this, the S-layer was also suggested to function as osmoprotectant in *Halobacterium salinarum*, as reflected in higher osmo-sensitivity of protoplasts[25].

As a first barrier, the S-layer potentially shields the cell from the environment, as exemplified by standard transformation protocols for halophilic and methanogenic archaea, which require transient removal of the S-layer in order to facilitate DNA uptake[26]. Correspondingly, it represents a first docking site for a number of extracellular molecules such as nutrients, vesicles, cells, and viruses[3,5]. Indeed, only recently, S-layer glycoproteins of hyperthermophilic archaea have been shown to accumulate iron and sulfur from the environment[27,28]. Also, recent studies revealed S-layer glycosylation to be crucial for cell-to-cell interactions during the mating process of the archaeon *Haloferax*[29]

and comparative genomics suggest a within-niche variation of glycosylation enabling *Haloferax* to discriminate between such mating partners and also viruses[30]. Similarly, virus- host interaction between *Halorubrum* sp. PV6 and HRPV-1 was shown to be mediated by recognition of *N*-formyl-legionaminic, a sugar residue decorating both the virus capsid protein and the host's S-layer protein[31,32]. However, given the possibility that *N*-glycan 5-*N*-formyl-legionaminic moiety is shared by additional surface proteins, the essentiality and indispensability of the S-layer for virus binding is not fully resolved. Even though knowledge of the actual role of the S-layer in virus and vesicle interaction is surprisingly sparse, the S-layer counteracts *Sulfolobus* spindle-shaped virus 1 (SSV1) egress because it seemed disrupted upon virus exit[33]. Taken together, the specific functions of the S-layer in archaea are still very little understood, and investigations of more S-layer depleted mutants are needed to clearly substantiate its role in the cell.

We have recently established a CRISPR-based post-transcriptional silencing technique for targeted gene knockdown in the hyperthermophilic archaeon *Sulfolobus solfataricus*[34,35]. The technique is based on the RNA-degrading CRISPR type III system, natively encoded in *Sulfolobus* species, which is reprogrammed to target host mRNA, instead of virus RNA. By transforming a miniCRISPR (miniCR) expressing synthetic small CRISPR RNAs (crRNAs) that matched a host mRNA at one or different positions (i.e., protospacers), we and others have efficiently and gradually silenced genes in different *Sulfolobus* species to different levels[34–37].

In this study, we applied the miniCR-based silencing technique to knockdown the S-layer gene *slaB* (SSOP1_0371), encoding the membrane-anchoring S-layer subunit in *S. solfataricus*. SlaB proteins are directly held in the membrane by a C-terminal helix domain and form stalk-like protrusions interacting with the outermost S-layer sheath composed of SlaA proteins (cf. Fig. 1a)[38]. The proteins constitute a p3 symmetry, and interact with other protein components of the cell envelope[6,39–41]. We show that 75% silencing of *slaB* expression leads to a partial loss and peeling off of S-layer units leading to a severely changed phenotype while higher silencing levels were not tolerated. Our study reveals important functions of the S-layer in *S. solfataricus*, and highlights the usefulness of the CRISPR-based silencing technology in archaea.

## Results

**Construction of miniCRISPR loci (miniCRs) for silencing.** For silencing of the *slaB* gene, three 37-bp target sites (i.e., protospacers: PS1, PS2, PS3) on the *slaB* mRNA were chosen such that their 3′ flanking sequence (PAS, protospacer adjacent sequence) matched at least 5 nt of the 5′ handle of the targeting crRNA expressed from a miniCR locus, in order to avoid CRISPR-mediated DNA degradation (Fig. 1a)[34]. Seven different miniCR constructs were designed matching the three different protospacers in the gene and carrying either single (SB2 and SB3, resp.) or multiple silencing spacers (SB23 and SB123, resp.) (Fig. 1a). In addition, two miniCRs carrying the targeting spacer SB3 two times (SB3×2) and six times (SB3×6), respectively, were designed with the attempt to yield stronger silencing by increasing the dosage of targeting crRNAs (Fig. 1a). Furthermore, a control construct "Ctrl" carrying the miniCR backbone only with two non-binding spacers, was designed. MiniCRs were assembled and inserted into the SSV1 virus-based shuttle vector pDEST-MJ[34], generating pDEST-SB vectors. To avoid any unforeseen vector-specific site effects and to increase the robustness of our results, we decided to use a complementary approach by expressing all miniCRs additionally from the plasmid-based vector pIZ, that we have recently adapted from the pRN1-based plasmid pCMalLacS

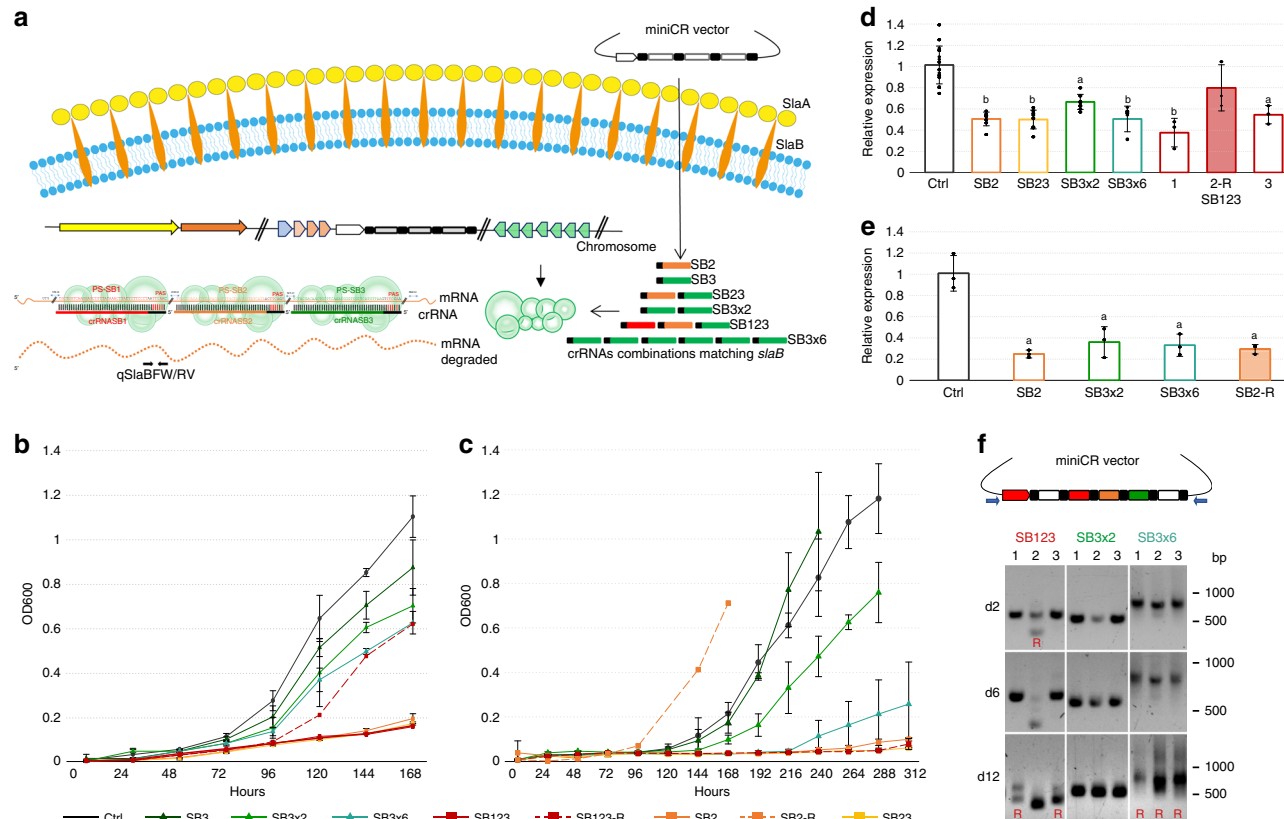

**Fig. 1** Silencing of *slaB* in *S. solfataricus*. **a** Schematic representation of the type III silencing pathway used for *slaB* knockdown. CrRNA assortments expressed from different miniCRISPRs in trans target the *slaB* mRNA at three possible positions (i.e., protospacer PS1, 2 or 3), each carrying a 3′ protospacer adjacent sequence (PAS). Protospacer positions (measured from start codon) and RT-qPCR—primer binding site are indicated (qSlaBFW and qSlaBRV). **b** Optical density increase of pDEST-SB-silenced cultures and pDEST-Ctrl control cultures (without targeting spacer). SB123-R represents a reverted (R) culture, which has lost the targeting spacer (cf. Fig. 1d and f). Dark gray (circle): control; dark green (triangle): SB3; green (triangle): SB3×2; petrol (triangle): SB3×6; red dash (rectangle): SB123 replicates; red dashed (rectangle) SB123-R; orange (rectangle): SB2; yellow (rectangle): SB23. Error bars, mean ± SD ($n = 3$, i.e., three different plaques picked and inoculated into liquid medium). **c** Optical density increase of pIZ-SB-silenced primary cultures and pIZ-Ctrl control cultures. Dashed orange line (SB2-R) represents a reverted (R) miniCR-SB2 culture, which arose by mutation of the targeting spacer upon culture transfer (cf. Fig. 1e). Color code/line markers and SD are defined as in **b**. **d** RT-qPCR on total RNA extracts of pDEST-SB-silenced cultures harvested at $OD_{600} = 0.1$. *SlaB* mRNA was quantified by qSlaBFW and qSlaBRV primers relative to the reference gene SSOP1_3283 (glyceraldehyde-3-phosphate dehydrogenase); Filled bars = reverted culture (R) (cf. 1c); Lower case letters indicate significant differences to Ctrl (two-tailed $t$ test, $n ≥ 3$, $p ≤ 0.007$). Error bars, mean ± SD (at least three biol. and tech. replicates). **e** RT-qPCR on total RNA extracts of pIZ-SB-silenced cultures at $OD_{600} = 0.1$. *SlaB* mRNA was quantified by qSlaBFW and qSlaBRV primers relative to the reference gene SSOP1_3283 (gapN-3); Filled bars = reverted culture (R); Lower case letters indicate significant differences to Ctrl (two-tailed $t$ test, $n = 3$, $p ≤ 0.0137$); Error bars, mean ± SD (three biol. replicates). **f** PCR products amplified by virus vector-backbone primers (blue) of pDEST–SB123, SB3×2, and SB3×6 transformants sampled at 2, 6, and 12 days after plaque inoculation, respectively. Three biol. replicates shown (1,2,3). Double bands and smears indicate loss of targeting spacers. Positions of DNA ladder are indicated (bp). Source data are provided[70]

used for transformation of *Sulfolobus acidocaldarius*[42,43]. Cells transfected with the virus-derived vectors (pDEST-SB2, -SB3×6, -SB23, and -SB123) produced plaques that were picked and grown as single transfectants for further analyses. For pIZ-plasmid transformations, single colonies harboring an intact miniCR, were only obtained from controls (Ctrl) or from complementation constructs (see below), never from cells transformed with silencing miniCRs only (Supplementary Fig. 1). Therefore, unless otherwise stated, pIZ-plasmid transformants carrying silencing miniCR obtained from primary cultures were used for further phenotypic studies.

**Growth defect in silenced cultures and recovery upon spacer loss.** Growth retardation of all transformants carrying a silencing miniCR was observed in liquid cultures (Fig. 1b, c). pDEST-SB virus vector transfectants harboring SB2, SB23, or SB123 were

most affected, reaching an $OD_{600}$ of 0.1 only after ~ 7 days in culture (Fig. 1b). In pIZ-plasmid transformants, SB123 and SB23 were instable, i.e., plasmids were repeatedly lost after an incubation time of ~ 7 days and cultures did not show any growth (Fig. 1c, Supplementary Fig. 2). In all transformants, growth retardation was more pronounced with the multiplex version SB3×6 compared to the corresponding single or double spacer constructs SB3 and SB3×2, respectively (Fig. 1b, c). All growth effects seemed construct-specific as similar growth patterns were repeatedly found in independently transformed cultures and regardless of the vector backbone being a virus or a plasmid (Fig. 1b, c). Growth recovery was observed in some cultures carrying miniCRs with strong silencing effects (SB123, SB23, SB2, SB3×6) upon longer incubation or transfer to fresh medium (examples in Fig. 1b, c dotted lines). These cultures had partly lost or mutated the *slaB*-targeting spacers of the miniCR locus (Fig. 1f). Loss of the targeting spacers in a subset of the

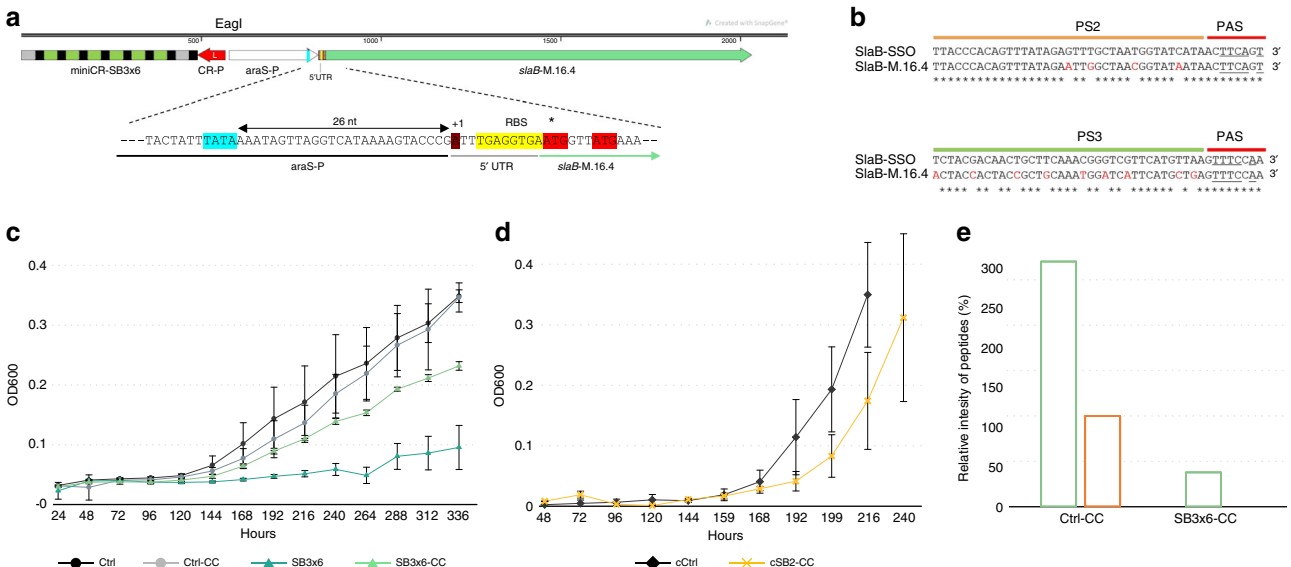

**Fig. 2** Complementation of *slaB*. **a** Fusion product of SB3×6 miniCR (CRISPR-promoter, red) and the complementation cassette: arabinose-inducible promoter: araS-P (TATA box, blue), 5′-UTR of *slaB S. islandicus* M.16.4 (M164_1762) including the putative TSS (dark red), a native ribosomal binding site (RBS, yellow) and two potential translation starts (ATG, red). **b** Nucleotide sequence alignment of *S. solfataricus* protospacer 2 and 3 (*slaB*-SSO) and respective regions on *S. islandicus* M.16.4 *slaB* gene (*slaB*-M.16.4). Asterisks: matching nucleotide; red: mismatches; underlined: PAS-5′ crRNA handle complementarity. **c** Optical density increase (arabinose supplemented medium) of pIZ-plasmid primary cultures carrying silencing constructs, control constructs (Ctrl) and complementation constructs (-CC), respectively. Dark gray (circle): control; light gray (circle): control-CC; petrol (triangle): SB3×6; light petrol (triangle): SB3×6-CC. Error bars, mean ± SD (three biol. replicates). **d** Optical density increase (arabinose supplemented medium) of pIZ-plasmid cultures recovered from single colonies (prefix "c") control and complementation constructs. Dark gray (rectangle): control; yellow (x): SB2-CC. Error bars, mean ± SD (three biol. replicates). **e** Relative intensity values (MaxQuant) of SlaB peptides of *S. solfataricus* and *S. islandicus* detected by mass spectrometry in Ctrl-CC and SB3×6-CC complemented cultures, respectively. Bars in the graph represent the samples in the following order from left to right (in the different cultures): *S. islandicus* SlaB (light green), *S. solfataricus* SlaB (brown). Y-axis shows intensity values normalized to Ctrl-CC SlaB-SSO in percent. Source data are provided[70]

population was indicated by multiple bands visible in gel electrophoresis of culture PCR products (Fig. 1f d2 and d6), which were identified by sequencing as fully intact and reduced versions of the originally supplied miniCR. Upon further incubation of those mixed cultures, bands corresponding to the intact miniCR disappeared completely (Fig. 1f d12). Spacer loss in cultures with pDEST-viral vectors preferentially seemed to happen precisely at repeat elements, probably through recombination events, leaving the vector including the CRISPR backbone intact. Such cultures are referred to as "revertants" (R), as the expression of *slaB* reverted to control-like levels (seen by RT-qPCR, Fig. 1d) apparently owing to the destroyed or reduced miniCR. Importantly, SB3 and SB3×2 miniCR constructs and control Ctrl were maintained stably in the culture. To analyze whether growth retardation was indeed paralleled by a reduced amount of *slaB* mRNA, RT-qPCR with specific primers was performed on total RNA extracts of the transformants sampled at an $OD_{600} = 0.1$ (Fig. 1a, Supplementary Table 1) and quantified in relation to the genes SSOP1_3283 (Fig. 1d, e) and the 16S rRNA (Supplementary Fig. 3) used as reference. Figure 1d shows significant silencing of all pDEST-virus vector transfectants in comparison with the Ctrl control cultures. In line with the observed growth retardations, the average silencing effect was ~ 50% for SB2, SB23, and intact SB123 cultures and was lower (~ 35%) in SB3×2 cultures. As expected, reverted culture SB123-2R, which recovered from growth owing to the loss of the targeting spacers (Fig. 1d, upper panel) did not show any significant reduction of *slaB* mRNA any more (Fig. 1d, filled bar). In pIZ-plasmid transformants, *slaB* silencing was assessed only for SB2, SB3×2, and SB3×6 samples, as the strongest miniCRs SB23 and SB123, were not stably maintained in culture (see above). The silencing trend was similar to the one observable for the pDEST-SB transfectants, however

showed an even stronger maximum knockdown level of ~ 75% with miniCR-SB2 and SB3×6 (Fig. 1e). Reverted culture SB2-R represented a mixed culture of silenced and escape mutants carrying mutations in the SB2 miniCR spacer, which accumulated after transfer to fresh medium. However, even in the partly reverted cultures, silencing seemed to be still prevalent at the timepoint of sampling as observed in plasmid culture SB2-R (Fig. 1e).

**Complementation with *S. islandicus* SlaB recovers phenotype.** To analyse whether the silencing effect could be rescued by the expression of a heterologous SlaB protein in trans, the *slaB* gene of the close relative *S. islandicus* (M.16.4_1762) was expressed together with the miniCR from the same vector (Fig. 2a). Among known *Sulfolobus* species, the SlaB of *S. islandicus* strain M.16.4 (i.e., SlaB-M.16.4) seemed best suited for complementation, as it showed the highest amino-acid identity (88% via BLAST) to SlaB of *S. solfataricus* (i.e., SlaB-SSO), whereas still containing sufficient divergence on the nucleic acid level (84% identity via BLAST). Dissimilarity on DNA level was necessary in protospacer regions to avoid targeting of the complementing *slaB* mRNA by the simultaneously expressed *slaB*-SSO-specific crRNAs (Fig. 2b). The *slaB*-M.16.4 gene (including 5′-UTR) was cloned downstream of an arabinose-inducible promoter and inserted into SB2, SB3×6, and Ctrl miniCR vectors (virus- and plasmid-based), resulting in Complementation Constructs (-CC) (Fig. 2a). Growth of complemented cultures (SB3×6-CC) has significantly recovered in comparison with the strong growth retardation observable for the non-complemented SB3×6 transformants cultivated in parallel (Fig. 2c, Supplementary Fig. 4). Increased cell fitness of the complemented cultures was further reflected by

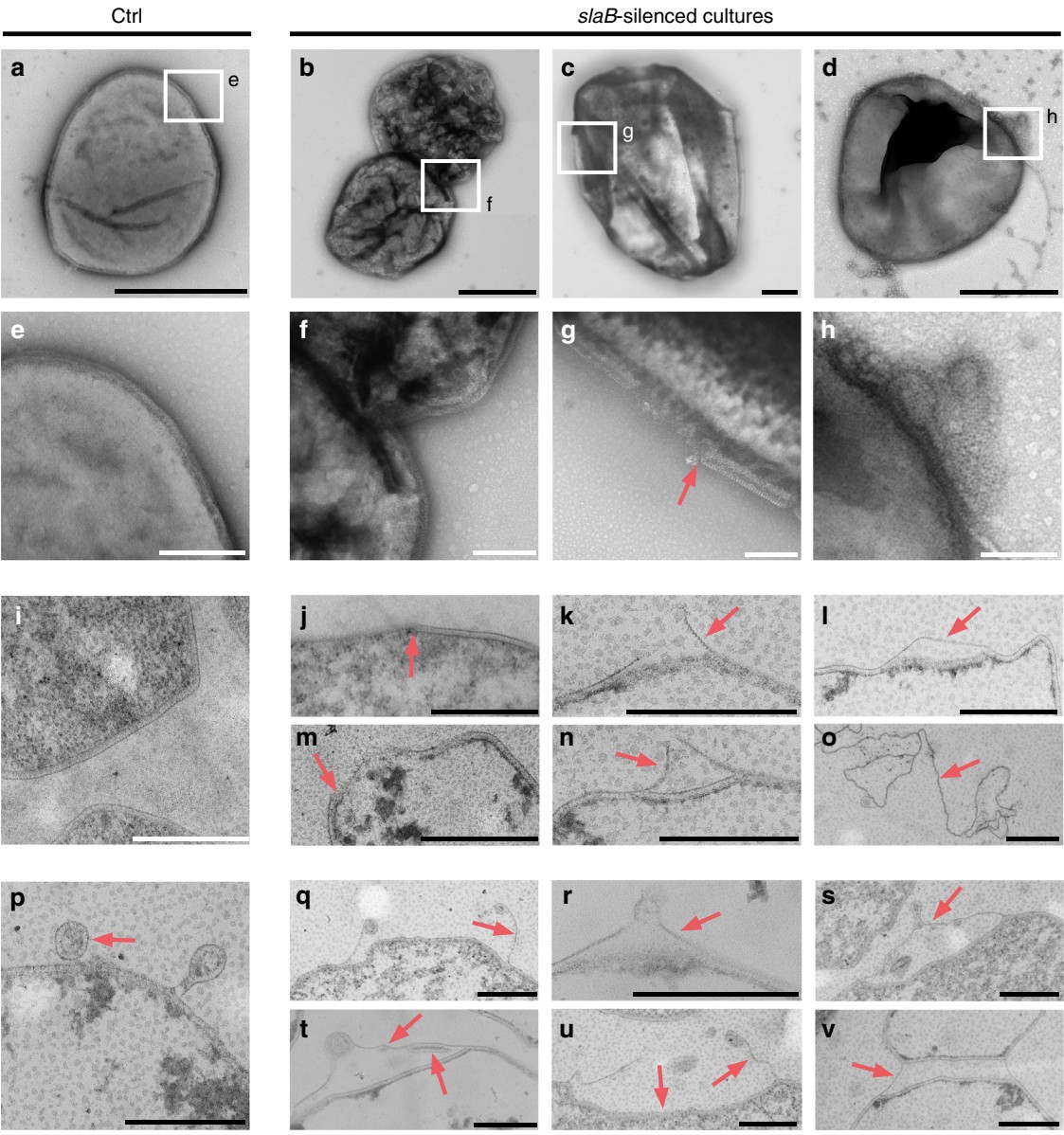

**Fig. 3** Electron micrographs of silenced and control (Ctrl) cultures. All cultures (pIZ-transformants) were harvested at $OD_{600} = 0.1$. **a–h** Transmission electron micrographs of negatively stained (2% Uranyl acetate) cells of Ctrl in **a** and SB3×6 cultures in **b–d** with indicated areas magnified in **e–h**. Scale bars: a–d: 1 µm; e–h: 200 nm. **i–o** Envelope morphologies of Ctrl in **i** and *slaB*-silenced SB3×6 and SB2 cultures in **j–o** visualized by transmission electron microscopy of thin sections (contrasted with Uranyl acetate 2% and Lead citrate 2%); Scale bar: 500 nm. **p–v** Vesicle secretion in Ctrl in **p** and *slaB*-silenced SB3×6 and SB2 cells in **q–v** visualized by transmission electron microscopy of thin sections (contrasted with Uranyl acetate 2% and Lead citrate 2%). Red arrows indicate S-layer. Scale bar: 500 nm

the ability of CC plasmid transformants to form single colonies on plates (Supplementary Fig. 1). Moreover, single colonies isolated from SB2-CC transformants showed similar growth rates as Ctrl colonies when transferred to liquid medium (Fig. 2d). To verify expression of SlaB-M.16.4, a band of ~ 40 kDa of complemented 3×6-CC and Ctrl-CC cultures was recovered from SDS-gels and analyzed by mass spectrometry. SlaB-M.16.4 was found in high quantities (~ $10^6$ intensity, MaxQuant) in both, Ctrl-CC and SB3×6-CC cultures. Data are represented relative to SlaB-SSO intensities of Ctrl-CC samples in Fig. 2e. Notably, SlaB-SSO was found in Ctrl-CC cultures only, substantiating silencing of the native *slaB* in SB3×6-CC on protein level.

**Silencing of *slaB* affects cell wall integrity and cell shape.** Electron micrographs of negatively stained cells revealed

heterogeneous populations in the *slaB*-silenced cultures compared with the homogenous phenotypes found in Ctrl (Fig. 3a). The majority of cells in silenced cultures showed high degrees of surface irregularities (Fig. 3b, f and Supplementary Fig. 5b–d). In addition, a subpopulation of cells was missing fragments of S-layer (Fig. 3c, g) or showed detached S-layer sheets often associated with egressing vesicles (Fig. 3d, h). The prevalence of irregular cells in silenced cultures was also reflected in flow cytometry (FC, Supplementary Fig. 6).

Thin sections of Ctrl, SB2, and SB3×6 provided further insight into the cell morphology and cell wall structure of silenced cultures. Ctrl cells showed a homogenous phenotype with lobed cells and a prominent cell wall structure as typical for *Sulfolobus* (Fig. 3i). Contrarily, in silenced cultures, we repeatedly observed cells that were missing segments of S-layer (Fig. 3j, m) or cells

where the S-layer lattice appeared to be peeling off of cells (Fig. 3k, n) or was partly loosened (Fig. 3l). Such gaps and irregularities in the S-layer structure were detectable in ~ 8% of thin-sectioned cells and showed gap widths of varying sizes with an average of 359 μm ($SD \pm 271$ μm, $n = 55$, Supplementary Fig. 7). Considering that the observable surface area of thin sectioned cells only represents a tiny fraction of the entire cell surface, the population of affected cells is certainly much higher. Compared with Ctrl (Fig. 3p), similar irregularities in vesicle secretion were observed in silenced cultures, where vesicles appeared to pull off S-layer sheaths from the cell during budding (Fig. 3q–u and Supplementary Fig. 5f). Interestingly, S-layer was observed in surface areas of peeled-off S-layer sheaths (Fig. 3n and s–v). If S-layer morphogenesis in *S. solfataricus* is comparably fast as in bacteria (500 proteins per second per cell)[3], these could represent freshly incorporated S-layer proteins. Partially detached S-layer lattice also seemed to connect cells to subcellular structures reminiscent of large "bud-like" appendages (Fig. 3v and Supplementary Fig. 5e, f) recently described for *Sulfolobus islandicus*[44]. Occasionally, structures resembling S-layer lattice fragments that were completely disconnected from cells were also observed in the preparations (Fig. 3o), similar to those found when S-layer had been chemically stripped from *Sulfolobus acidocaldarius*[45].

The various microscopic observations provide evidence that the reduced SlaB level has a pronounced effect on the appearance and shape of the cells and on the cell envelope integrity.

**Reduced surface glycosylation in *slaB* knockdown cultures**. To analyze how far the reduction of the SlaB protein influenced the surface glycosylation of the cell, glycosylation profiles of all cultures were analyzed by lectin stains (WGA, Alexa Fluor 488 Conjugate, ThermoFisher Scientific) (Fig. 4, Supplementary Fig. 8, Supplementary Fig. 9). Compared to the Ctrl culture, 60% and 45% of cells in the silenced SB2 and SB3×6 cultures, respectively, showed decreased fluorescence, indicating a reduction of surface glycosylation (Fig. 4 "low"). The fluorescence profile of the complemented cells SB3×6-CC, SB2-CC, and control Ctrl-CC indicated a higher degree of surface glycosylation compared to silenced cultures and Ctrl cells. This shift was most likely caused by the integration of the *S. islandicus* SlaB, which has been predicted to contain more potential glycosylation sites than the *S. solfataricus* protein[39,46]. These observed changes in surface glycosylation suggest SlaB to be a decisive constituent of the continuous sugar coat of a *Sulfolobus* cell.

***SlaB* silencing leads to increased cell size**. Phase contrast and electron microscopy revealed dramatic cell size heterogeneity in silenced cultures exhibiting up to fourfold larger cells compared with Ctrl control cultures (Fig. 5a). When measuring cell sizes manually by analyzing phase contrast micrographs of fresh cultures ($n > 300$ for different biol. replicates), we found silenced cells to average at 3 μm, whereas the regular Ctrl cell size was 1.6 μm (Fig. 5b). Cell sizes of 65% of counted cells in *slaB*-silenced cultures were ranging between 2.5 and 8.5 μm, clearly exceeding the maximum cell size abundantly found in Ctrl cultures (> 90% between 1.5 and 2 μm) (Fig. 5b). This trend was reproduced by FC ($n = 10,000$ per biol. replicate), where the FSC-histograms (FSC = forward scatter) were positively skewed in silenced cultures, reflecting the prevalence of larger cells (Fig. 5c). To relate abundances of the large cell fractions in the silenced culture to the Ctrl culture, the FSC-histogram of Ctrl cultures was divided into "small cells" (25% of culture), "average cells" (50% of culture), and "large cells" (25% of culture). Approximately 40% of SB3×6 and SB2 cells fell into the "large cell" section, whereas complemented cultures showed a significantly lower proportion of larger cells (28%), similar to Ctrl (Fig. 5c "large"). Enlarged cells occasionally appeared to have a reduced cytosol density in thin sections (Fig. 5a, lower right panel). However, as live-dead staining before thin sectioning showed that both large and regularly-sized cells were indeed viable and contained DNA (Supplementary Fig. 10), we conclude that cell wall depletion destabilized the cells to such a degree that they became more susceptible to damages during the preparation for electron microscopy, as previously observed in other S-layer mutants[47].

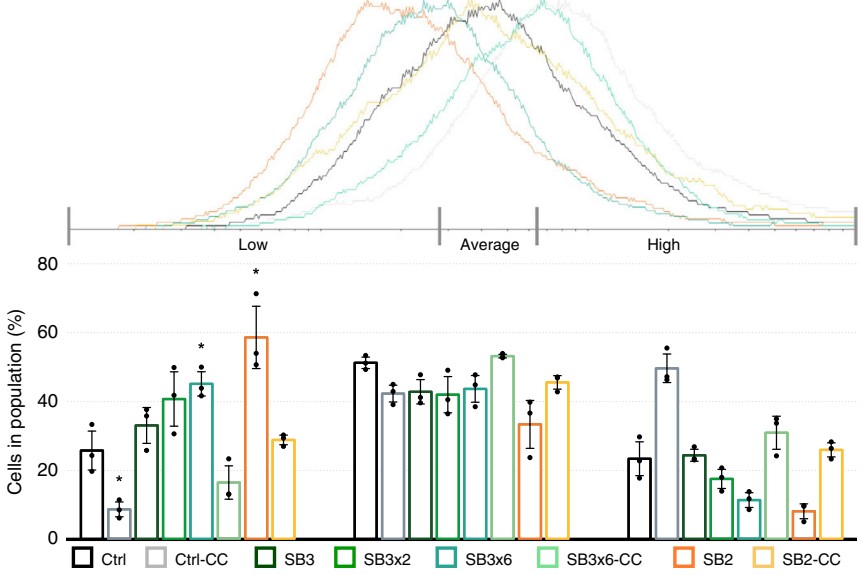

**Fig. 4** Surface glycosylation in silenced and control pIZ-plasmid cultures. FC-Histogram (fluorescent flow cytometry, created with *Flowing Software 2.5.1*, 3 biol. repl, $n = 10,000$ per replicate) of lectin stained (WGA-Alexa488) cultures harvested at $OD_{600} = 0.1$. Bar chart represents percent of cells in population showing low, average or high fluorescence (normalized to cell size). Sections were defined according to Ctrl histograms. Bars in the graph represent the samples in the following order from left to right (in every section): Control (dark gray); control-CC (light gray); SB3, (dark green); SB3×2 (green); SB3×6 (petrol); SB3×6-CC (light petrol); SB2 (orange); SB2-CC (yellow). Error bars, mean ± SD (three biol. repl.). Asterisks indicate significant differences to Ctrl (two-tailed *t* test, $n = 3$, $p \leq 0.037$). Source data are provided[70]

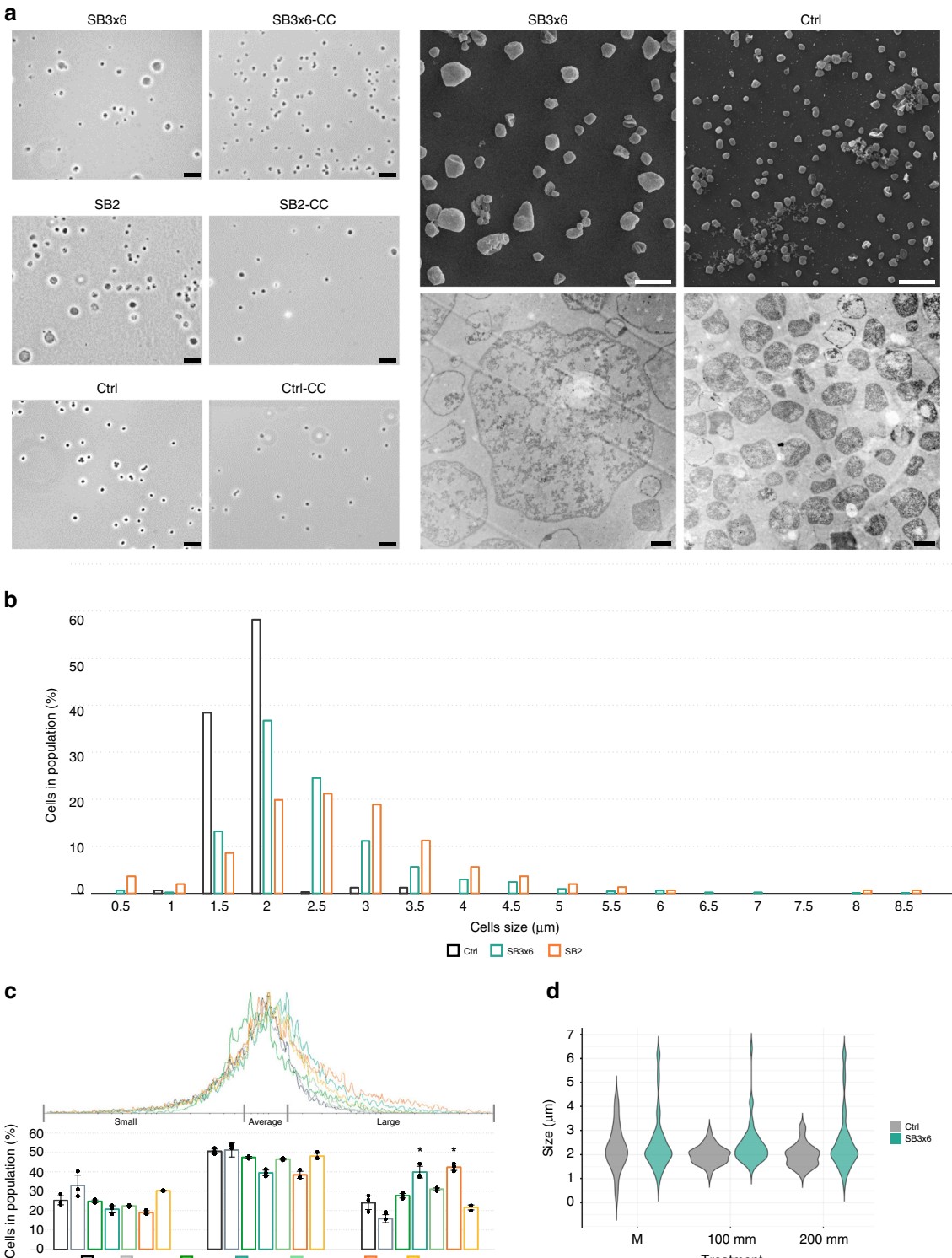

When silenced and control cultures were exposed to increasing NaCl concentrations, a general decrease of the average cell size was observed, which was slightly higher in SB3×6 (23% diameter decrease) than in Ctrl cultures (19% diameter decrease) (Fig. 5d). However, enlarged cells (7 μm) persisted over salt-treatments, indicating that they were stable upon hyperosmotic stress (Fig. 5d). Generally, as the observed differences between *slaB*-silenced and control cultures were relatively small, the S-layer of *Sulfolobus* might not play a major role in providing osmotic cell stability, at least not upon moderate hyperosmotic changes.

**SlaB knockdown affects cell division and chromosome copy numbers**. In order to investigate a possible correlation between the large cell phenotype and the measured growth retardation (Fig. 1b, c), fresh cells of silenced and control pIZ-plasmid cultures were harvested precisely at $OD_{600} = 0.1$ and cell numbers as well as average sizes were determined by FC (Fig. 6a). In total, $1.12 \times 10^7$ and $1.59 \times 10^7$ cells per ml were counted on average in SB2 and SB3×6-silenced cultures, respectively, which was around half of the cell numbers measured in Ctrl cultures ($3 \times 10^7$ cells per ml) (Fig. 6a). The lower cell count in knockdown cultures was

**Fig. 5** Cell size distribution in silenced and control pIZ-plasmid cultures. **a** Left: phase contrast micrographs of Ctrl control, SB3×6 and SB2 *slaB*-silenced (left) and complemented cultures (-CC) (right). Scale bar: 10 μm. Right upper panel: scanning electron micrographs (gold coated) of *slaB*-silenced SB3×6 and Ctrl; scale bar: 5 μm (magnification × 2700). Right lower panel: Transmission Electron micrographs (magnification: × 6000) of thin sections (contrasted with Uranyl acetate 2%, and Lead citrate 2%) of *slaB*-silenced SB3×6 (left) and control Ctrl cultures (right); scale bar: 1 μm. **b** Cell size distribution of Ctrl, SB3×6, and SB2 cultures ($OD_{600} = 0.1$). Bars in the graph represent the samples in the following order from left to right: Control (dark gray); SB3×6 (petrol); SB2 (orange). Cell sizes were measured using MicrobeJ software on phase contrast micrographs ($n > 300$, different biol. replicates). **c** FSC-histogram (Forward scatter, flow cytometry, created with *Flowing Software 2.5.1*) of silenced and control cultures sampled at $OD_{600} = 0.1$ ($n = 10,000$ per biol. replicate). Bar chart represents percentage of cells in population exhibiting "small", "average" and "large" size (size sections defined according to Ctrl histograms). Bars in the graph represent the samples in the following order from left to right (in every section): Control (dark gray); control-CC (light gray); SB3×2 (green); SB3×6 (petrol); SB3×6-CC (light petrol); SB2 (orange); SB2-CC (yellow). Error bars, mean ± SD (three biol. replicates). Asterisks indicate significant differences to Ctrl (two-tailed *t* test, $n = 3$, $p \leq 0.009$) **d** Salt-stress experiment. Violin plot depicts decrease and increase of abundances of specific cell sizes of untreated (M) pIZ-plasmid SB3×6 and Ctrl cultures ($n = 50$ cells per replicate and condition), and upon 100 mM and 200 mM salt treatment (treatment upon salt addition = 45 min, 77 °C, agitating), respectively. Plots represent the samples in the following order from left to right (in each treatment): Control (gray); SB3×6 (petrol). Cell size differences were measured as in **b**). Source data are provided[70]

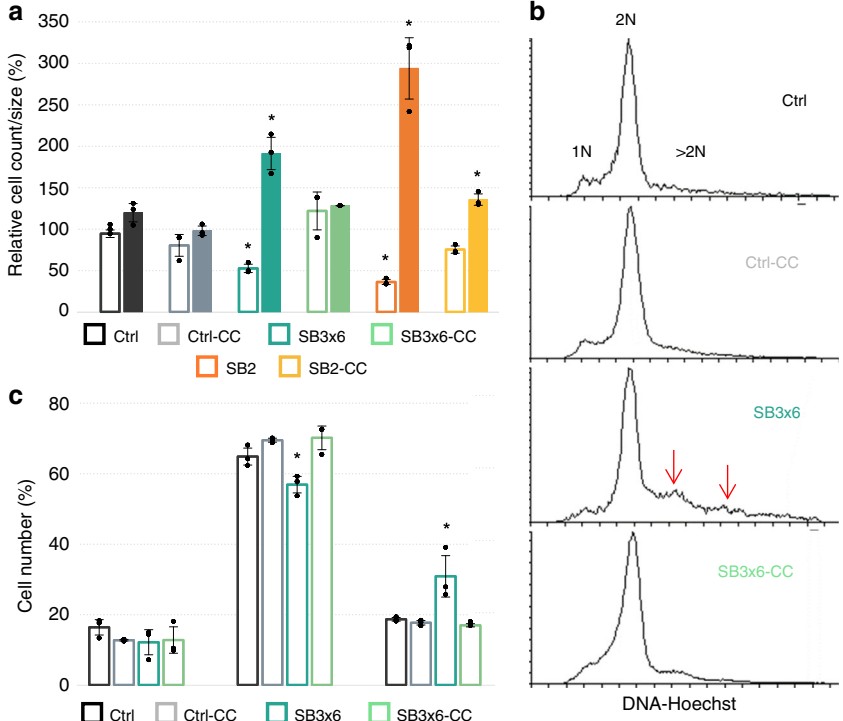

**Fig. 6** Correlation of cell count, size, and DNA content in silenced and control cultures of pIZ-cultures. **a** Differences in cell counts (cells per ml) determined by flow cytometry (calculated from flow rate) and cell size (FSC, $n = 10,000$ per biol. repl.) of same cultures harvested at precisely $OD_{600} = 0.1$ and normalized to Ctrl control. Cell counts: empty bars, cell size: filled bars. Bars in the graph represent the samples in the following order from left to right (for both conditions): control (dark gray); control-CC (light gray); SB3×6 (petrol); SB3×6-CC (light petrol); SB2 (orange); SB2-CC (yellow). Error bars, mean ± SD (three biol. repl.). Asterisks indicate significant differences to Ctrl (two-tailed *t* test, $n \geq 3$, $p \leq 0.018$). **b** Fluorescence histograms of Ctrl, Ctrl-CC, SB3×6, SB3×6-CC pIZ-plasmid cultures harvested at $OD_{600} = 0.1$ and stained with Hoechst. Peaks represent cell abundances containing one chromosome (1 N), two chromosomes (2 N), and more than two chromosomes (>2 N) in respective cultures (created with *Flowing Software 2.5.1*). **c** Percentage of cells containing defined chromosome numbers in each culture (as in **b**): 1 N (left bars), 2 N (middle bars), >2 N (right bars). Bars in the graph represent the samples in the following order from left to right (in every section): Control (dark gray); control-CC (light gray); SB3×6 (petrol); SB3×6-CC (light petrol). Error bars, mean ± *SD* (three biol. repl.). Asterisks indicate significant differences to Ctrl (two-tailed *t* test, $n \geq 3$, $p \leq 0.049$). Source data are provided[70]

paralleled by an increased average cell size (FSC, $n = 10,000$), which was over one third (SB3×6) larger or even over double (SB2) the size of a regular Ctrl cell (Fig. 6a), reflecting size ratios shown above (cf. Fig. 5b). Notably, cell counts, and sizes of complemented cultures were in the range of the Ctrl controls (Fig. 6a). These results indicate that the moderate optical density increase measured in silenced cultures (Fig. 1b, c) reflected growth in cell width rather than cell division. Furthermore, *slaB* silencing seemed to affect chromosome copy numbers, as *slaB*-silenced cultures carried a significant proportion of cells that

contained more than two chromosomes (>2 N) (Fig. 6c). This proportion was visible as fluorescent signals in areas designating 3 N and 4 N ploidy in FC-Hoechst histograms of SB3×6 cultures (Fig. 6b, Supplementary Fig. 11), which differed from described 1N–2N patterns of exponentially growing *Sulfolobus*[48]. Contrarily, histograms of both control and complemented cultures exhibited 1N–2N patterns only, without distinct >2 N peaks (Fig. 6b). Increased cell size paralleled by increased DNA content in *slaB*-silenced cells indicated a collapse of cell partitioning and suggested a significant role of an intact S-layer in cell division.

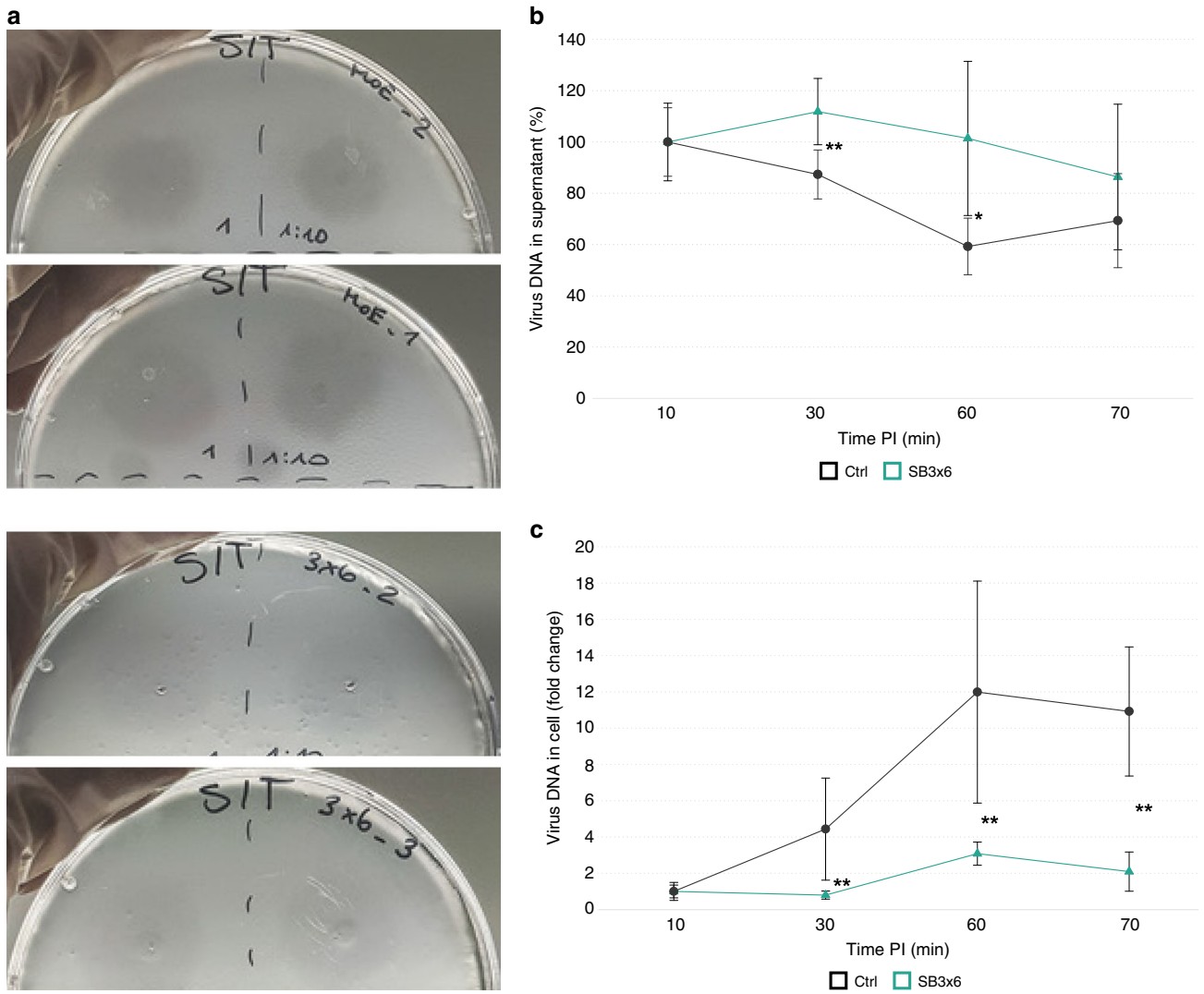

**Fig. 7** Infection assays on *slaB*-silenced and control cultures. **a** Spotting assay using different dilutions of purified SSV1 virions on cells transformed with pIZ-Ctrl (upper panels) and pIZ-SB3×6 (lower panels), respectively. Two biological replicates presented (the same virion sample was used for the different overlays). **b** Relative amounts of extracellular virus DNA measured at certain timepoints postinfection (PI) in supernatants of pIZ-SB3×6 (petrol, triangle) and pIZ-Ctrl cultures (dark gray, circles), which have been challenged with SSV1 virions at MOI 0.1. Virus copies were determined by qPCR using D-291 FW/RV primers (Supplementary Table 1) and normalized to cell count in the individual samples and to the initial virus added. Percentage in relation to timepoint 10 min PI is shown. Error bars, mean ± SD (three biol. and techn. replicates). Asterisks indicate significant differences to Ctrl (two-tailed *t* test, $n \geq 3$, $p \leq 0.0105$). **c** Fold change of increase of virus DNA inside cells in same samples as in **b**. Virus DNA measurement was conducted as in **b**; error bars are defined as in **b**. Asterisks indicate significant differences to Ctrl (two-tailed *t* test, $n \geq 3$, $p \leq 0.0083$). Source data are provided[70]

**SlaB silencing impairs infection by SSV1 virus.** To investigate whether *slaB* knockdown affected virus–host interactions, infection assays with SSV1 virions were performed. First, spotting assays using different dilutions of SSV1 virion solutions on an overlay of concentrated SB3×6 and Ctrl pIZ-plasmid-transformants were conducted. Plaques formed after 48 h of incubation were of the same size but considerably more turbid on SB3×6 overlays than on Ctrl, indicating that *slaB*-silenced cells were less sensitive to virus infection (Fig. 7a). Noteworthy, plaque morphologies of control cultures were comparable to those generated on *Sulfolobus* wild-type lawns[49], excluding any influence of the plasmid backbone on plaque formation. To assess the efficiency of virus uptake over time, we mixed SSV1 virions with SB3×6 and Ctrl pIZ-plasmid—transformed cells at a multiplicity of infection (MOI) of 0.1 and incubated the suspensions at 77 °C under agitation. Samples were taken at regular intervals up to 70 min postinfection (PI) and virus DNA in supernatant and cell

pellets was quantified by qPCR. Data were normalized to cell count and to initially added virus. Owing to unequal dispersal of virions in the culture, timepoints before 10 min PI were not considered and virus uptake was thus compared with 10 min PI (T10). With 87% of extracellular virus DNA measured, significantly less virus DNA was detected in control cultures at 30 min PI compared with 10 min PI (Fig. 7b). The number of extracellular virus further decreased in Ctrl samples at 60 min PI, where only 59% of the initially measured virus DNA could be detected (Fig. 7b). Contrarily, no significant reduction of extracellular virus DNA compared with T10 was observed for neither of the timepoints PI in SB3×6 cultures (Fig. 7b). Complementarily, intracellular virus copies increased fourfold at 30 min PI in Ctrl cultures, whereas almost no increase was detectable for SB3×6 (Fig. 7c). Only after 60 min PI, SB3×6 showed an increase in intracellular virus DNA, which, however, was still significantly lower than the 11-fold increase measured for the same timepoint

**Table 1 Relative amounts of SSV1 virus DNA of silenced and control cultures infected at different MOIs**

| MOI, n | V-DNA found in | PI (min) | SB3×6 | Ctrl | Sig. |
|---|---|---|---|---|---|
| MOI: 0.1 n = 3 | Supernatant | 30 | 112 (±13) | 87 (±10) | ** |
| | | 60 | 101 (±30) | **59**[a] (±11) | * |
| | Cell | 30 | 0.80 (±0.22) | **4.44** (±2.82) | ** |
| | | 60 | **3.08** (±0.64) | **11.99** (±6.12) | ** |
| MOI: 5 n = 3 | Supernatant | 35 | 75 (±20) | 43 (±11) | |
| | | 50 | **43** (±3) | **24** (±8) | * |
| | Cell | 35 | 0.97 (±0.17) | **1.60** (±0.28) | *** |
| | | 50 | 0.97 (±0.38) | **1.45** (±0.30) | * |
| MOI: >100 n = 3 | Supernatant | 30 | **71** (±14) | 59 (±6) | * |
| | | 60 | **63** (±6) | 43 (±7) | *** |
| | Cell | 30 | **1.66** (±0.23) | **1.85** (±0.28) | |
| | | 60 | **2.14** (±0.13) | **2.00** (±0.19) | |

Extracellular (supernatant) and intracellular (cell) virus amounts of same cultures were measured in relation to timepoint 10 min PI as percentage and fold change, respectively. Measurements of the same cultures (three biol. repl. each) at two different timepoints postinfection (PI) are presented. Significant differences (column Sig.) between SB3×6 and Ctrl at the same timepoint are indicated with asterisks, where $p \leq 0.05$ (*), $p \leq 0.01$ (**), $p \leq 0.001$ (***) (two-tailed t test, $n \geq 3$)[70]
Bold values indicate significant differences of respective timepoints to timepoint 10 min PI in each individual culture ($p \leq 0.009$ for extracellular; $p \leq 0.05$ for intracellular)

in Ctrl cultures (Fig. 7c, Table 1). At 70 min PI, neither extracellular nor intracellular virus DNA levels changed in SB3×6, respectively (Fig. 7b, c). Contrarily, extracellular virus DNA of control cultures tendentially increased again at 70 PI, indicative for a release of freshly produced virions (Fig. 7b). These data show that the efficiency of virus uptake, and therefore, virus susceptibility is significantly higher in Ctrl cultures than in the S-layer-depleted cells at every timepoint examined (Table 1). To investigate if the same effect can be seen at higher virus titers, this experiment was repeated at MOI 5 and MOI > 100 independently, infecting three biological replicates each (Table 1, Supplementary Fig. 12). All infected cultures followed the similar trends as observed for MOI 3, showing a more-efficient decrease of supernatant virus DNA in control cultures, which in all cases, significantly differed from SB3×6, at least at 60 min PI (Table 1). Noteworthy, complementary results were obtained using plaque assays to assess the number of the extracellular virions, as performed for MOI 5, proving the reproducibility of the results using a different experimental approach (Table 1, Supplementary Fig. 12). Apart from MOI > 100, intracellular virus DNA increased over time and was significantly higher in control samples than in SB3×6 at all timepoints analyzed (Table 1). The high virus titer at MOI > 100 probably complicates the identification of smaller differences.

## Discussion

In this study, we were able to silence the *slaB* gene encoding the membrane-anchoring S-layer protein, up to 75% using the recently established CRISPR type III-mediated silencing technology[34,35]. Synthetic crRNAs matching the *slaB* mRNA at different positions were expressed in trans from a miniCR array carried either with a plasmid or a virus vector. Mechanistically, the expressed crRNAs incorporate into a multi-subunit type III complex each and guide it to the complementary protospacers (target regions) on the *slaB* mRNA, which is subsequently cleaved by the ribonucleolytic action of the Cmr4 subunits[50]. Apart from disabling DNA degradation[51], PAS-handle binding as used in our experiments, also inhibits the synthesis of cyclic oligoadenylate by the type III subunit Cas10, which was recently shown to induce unspecific RNA shredding by activation of HEPN family ribonucleases[52]. Therefore, we can preclude any unspecific RNA or DNA degradation when targeting the *slaB* mRNA using the CRISPR type III-miniCR combo. In eukaryotic RNAi experiments, multiple small RNAs targeting the same gene are used to reject phenotypical changes caused by unspecific off-targeting[53]. Similarly, in our study, two different crRNA (SB2 and SB3)

species caused the same phenotypical changes (to same extents when multiplexed) in *slaB* knockdown cultures, which substantiates the targeting specificity. Furthermore, simultaneous expression of a SlaB from *Sulfolobus islandicus* M.16.4 in both SB2 and SB3×6-silencing experiments rescued silenced phenotypes. This again verified the specificity of the crRNAs and demonstrated that the observed phenotypes indeed resulted from reduced SlaB.

SlaB depletion led to an apparent decrease in binding affinity of the whole S-layer lattice to the cell as seen by electron microscopy showing cells with a discontinuous S-layer coat and with S-layer sheaths peeling off the cell. The protein composition of the detached S-layer sheaths has not been determined in this study. However, SlaA subunits are thought to interact with each other based on their presence as dimers in SDS gels[38]. Moreover, preliminary data of our group provide evidence that SlaA can indeed self-assemble in the absence of SlaB. Thus, the detached sheaths could potentially be formed solely by SlaA proteins, suggesting SlaA subunits to tightly bind to each other but not to the membrane, unless efficiently anchored by the help of SlaB. Detachment of the S-layer sheaths from the *slaB*-depleted cell envelope was increasingly apparent in cells that showed vesicle protrusions, where the budding vesicle appeared to "pull" the S-layer lattice off the cell wall, a phenomenon which has never been reported for wild-type cells[54,55]. We therefore suggest that a SlaB-depleted S-layer also affects vesicle secretion and coating, as those vesicle poles facing the cell did not seem to be properly covered with the S-layer (cf. Fig. 3r, 3t). The destabilized S-layer lattice had a dramatic impact on cell shape, cell size, and glycosylation. Negative stains revealed cells in *slaB*-depleted cultures to be more interfolded than control cells, indicating a role in shape maintenance. This complies with earlier observations that lattice folds (i.e., missing S-layer parts) promote the invagination of the membrane, potentially leading to cell shape alterations[16]. Furthermore, the substantial reduction of modifying surface sugar residues in *slaB*-silenced cultures illustrates that the S-layer majorly contributes to the cell's sugar coat. This coincides with the finding of 31 potential, and five modified *N*-glycosylation sites of SlaA and SlaB proteins, respectively[39,56]. Envelope glycosylation was reported to heavily affect surface properties and motility in mutants of both bacteria and archaea[57,58] and was speculated to be involved in adaptation to changing environments[5]. Thus, our findings emphasize the importance of a continuous S-layer for the preservation of the physiologically significant sugar envelope.

Silenced cultures revealed an up to fivefold-increase in cell size, as seen by microscopy and flow cytometry. There is another study

describing size increase of an S-layer stripped archaeon. *Methanobacterium thermoautotrophicum* protoplasts underwent massive cell shape alterations from rod-shaped to spherical with spheres further increasing in diameter over time. However, the reason for the size increase was not addressed, and *Methanobacterium* differs from *Sulfolobus* by the presence of a pseudomurein layer acting as additional cell wall[59]. Notably, cell size increase was often observed in cell division impaired *Sulfolobus* cells[44,60–62] where catalytically inactive derivates of cell division (Cdv) proteins were overexpressed, or respective genes were disrupted, respectively. Interestingly, enlarged cells often showed elevated DNA content[60,61] which was accompanied by a growth defect and cell division failure[61]. This phenotype strongly resembles the *slaB*-silenced cells in our study, as we observed growth retardation and a significant lower cell count at same optical densities, indicating cellular width-growth instead of cell propagation. Strikingly, we also found a deviant accumulation of DNA copies affecting at least 30% of the *slaB*-depleted culture, implying that those cells replicated their DNA, yet failed in undergoing proper fission. Notably, cell sizes, growth, cell count, and DNA content of complemented cultures did not significantly differ from control cells, substantiating that these phenotypes were indeed caused by the reduced S-layer proteins. We therefore conclude that the dramatic size increase accompanied by elevated DNA copies, is caused by impaired cell division due to the depleted S-layer. It has been suggested earlier, that the role of the S-layer in cell division is crucial, potentially by driving membrane invagination through integration of newly synthesized S-layer units at the septum side[14–16,20]. S-layer depletion might destabilize such a process, yet the cell would mount for multiplication, which in turn would lead to accumulation of DNA and other cell components (i.e., lipids). As CdvA was shown to bind to the membrane[63], a direct interaction with the transmembrane SlaB protein could in theory be possible. Thus, impairment of cell division could also be caused by a disturbed regulation between S-layer and cell division machinery. The fact that S-layer depletion causes phenotypes reminiscent to those of cell division machinery mutants further corroborates an interaction between an intact S-layer and membrane-linked processes.

The attachment site and host cell entry of the widespread *Sulfolobus spindle*-shaped viruses has remained elusive up to now. Our finding that SlaB-depleted cultures were significantly less susceptible to SSV1 infection than the control cells, indicates an important role of the S-layer on the viral infection cycle. However, it remains to be investigated whether the S-layer functions directly as a virus receptor or secondarily contributes to virus attachment or entry. Contrarily to archaeal filamentous viruses, spindle-shaped viruses were not found attached to cellular appendages, but directly to the cell surface[64]. Also, remarkably high numbers of virions were often observed to attach to the same cell, indicating the cell surface receptor to be highly abundant[65,66]. Thus, as the S-layer accounts for the most abundant surface proteins, we consider it possible, that either SlaA or SlaB constitute the virus attachment site. Strain-specific glycan-modifications of the S-layer proteins could potentially serve as binding sites, as polysaccharides are common receptors for bacteriophages[67] and were suggested to play a role in virus susceptibility within the archaeal species *Halorubrum* sp. PV6[32]. The S-layer could also secondarily contribute to virus–host interactions, potentially by sustaining the actual surface receptor site. Indeed, the S-layer was found to stabilize other surface proteins, such as the Archaellum and the bindosome-system[40,41].

In summary, the strong phenotypical effects observed in this study infer that the function of SlaB as S-layer anchor, and the presence of the S-layer as a whole, is important for a proper life cycle of *S. solfataricus*. This is also reflected by the fact that a maximum silencing effect of 75% could be achieved, whereas stronger silencing constructs were not tolerated. In contrast, we have recently shown that 98% silencing of the non-essential α-amylase gene was possible without any host-induced alterations of the miniCR array[35]. Interestingly, recent studies in the relative *S. islandicus* M.16.4 showed that *slaB* gene deletion (or deletion of *slaA* or a third gene involved in S-layer formation) was possible without causing strong growth delay, even though the majority of the cells showed a partially peeled off (or even fully peeled off) S-layer[23,24]. In those studies, cultivation was performed under non-shaking conditions, and S-layer mutant cells were found clustered in aggregates[23]. Formation of cell clusters and biofilms in *Sulfolobus* is known to form as a protection mechanism upon stress and to be accompanied by changes in gene expression[68,69]. In our study, cultures were agitated throughout experiments and no cell aggregation was observed. Therefore, it is possible that the strong phenotypes observed in our study might be revealed owing to the harsher cultivation conditions, not promoting cell aggregation, which we believe better resemble the dynamic environment of the natural habitats of *S. solfataricus*, such as bubbling solfataric hot springs.

## Methods

**Design of genetic constructs**. Three different 37-bp target regions (i.e., proto-spacers, PS1–3) were selected on the *slaB* gene (SSOP1_0371) with the following coordinates on the *S. solfataricus* P1 genome (GenBank: LT549890.1): PS1: 338615–338651, PS2: 338870–338906, PS3: 339036–339072. All protospacers were chosen to be flanked by a PAS motif at the 3′-end[34]. MiniCR-spacer sequences SB1-3 were designed to be complementary to the respective protospacer on the *slaB* mRNA, when transcribed to crRNAs (CRISPR RNAs) (Supplementary Table 1). Spacers for miniCR constructs were assembled by fusion of primer pairs miniCR-SB1FW/RV for spacer SB1, miniCR-SB2FW/RV for spacer SB2 and miniCR-SB3FW/RV for spacer SB3, respectively, by PCR (Supplementary Table 1)[35]. Furthermore, fusion of primers Ctrl-FW, Ctrl-RV created the non-targeting miniCR array "Ctrl", carrying the miniCR backbone spacers D1 and D5 only[34]. Further assembly of the different spacers to single- and multiplex miniCR constructs were performed as described in ref. [35] with the difference that a pENTRY-TL (truncated leader) vector was used as template in the final PCR step, which carried a truncated version (77 nt-sequence upstream of the first repeat) of the native CRISPR locus D of *S. solfataricus* P2. Dosage constructs SB3 two and six times, respectively, were constructed by linearizing the pENTRY-SB3 vector using primer pair SlaB3_Lin_FW and SlaB3_Lin_RV within the spacer region. Low-cycle PCR using primer pairs SlaB3_duplex_FW/RV extended the linear fragment by one additional SB3-spacers, generating the SB3×2 array[35]. Concatemers of different length formed in PCRs using SlaB3_triplex_FW/RV owing to recombination events between homologous sequence stretches. The SB3×6 miniCR was therefore obtained by gel purification of a band at corresponding height (Monarch Gel purification kit, NEB), which was further verified by sequencing. MiniCRs and complementation cassettes (see below) were finally transferred from pENTRY-TL vectors onto the expression vectors pDEST-MJ[34] and pIZ-GW[43], respectively, by Gateway in vitro recombination (Invitrogen, Life technologies). pIZ- plasmids carry the *pyrEF* operon of *S. solfataricus* P2 for complementation of uracil auxotrophy and were shown to be stably maintained extrachromosomally as a single-copy plasmid in *S. solfataricus* P1[43].

For complementation constructs, the *slaB* gene of *S. islandicus* M.16.4 (M.16.4_1762, DNA, and *S. solfataricus* were aligned, which revealed M.16.4 *slaB* 4 and 10 mismatches in complementary regions of protospacers PS2 and PS3, respectively (cf. Fig. 2b). As mismatches were scattered throughout PS3, and PS2 exhibited two mismatches towards the 3′-end, which is considered important for target cleavage[50], *slaB*-M.16.4 was considered divergent enough to avoid cross-targeting of simultaneously expressed SB2 and SB3-crRNAs. For creation of complementation cassettes, first the araS promoter of *S. solfataricus* P2[34] was PCR-amplified (Phusion Polymerase, ThermoFisher, Scientific) using araFW_EagI and araOH_MSlaB_RV primers (Supplementary Table 1), which added an *Eag*I recognition site to the 5′ and an overhang consisting of the first 25 nt of the *slaB*-M.16.4 sequence (see below) to the 3′-end of the *araS* sequence, respectively. The sequence chosen for *slaB*-M.16.4 expression contained the *slaB*-M.16.4 coding sequence (CDS), as well as the 11 bp 5′-UTR containing a RBS[38], as well as the putative terminator sequence downstream of the CDS (genome coordinates: 1615528–1614238, GenBank: CP001402.1). This region was amplified (DNA extracted from *S. islandicus* M.16.4 obtained from the Rachel Whitaker Laboratory) using primers SlaBM164_FW and SlaBM164_SalI_RV, thereby adding a *Sal*I restriction site to its 3′-end (Supplementary Table 1). AraS and *slaB*-M.16.4 PCR products were gel-purified (Monarch Gel purification kit, NEB) and subsequently fused at the homologous overlap region using primers araFW_EagI and

SlaBM164_SalI_RV, in a touch-down PCR reaction (Supplementary Table 1). The thereby assembled complementation cassette was gel-purified, verified by sequencing and further cleaved by *Eag*I-HF (NEB) and SalI-HF (NEB) following manual's instructions. Analogously, respective pENTRY-miniCR-constructs (see above) were cleaved by the same enzymes, respectively, as a *Eag*I and a *Sal*I restriction site are sequentially located 5 bp upstream of the truncated leader promoter (cf. Fig. 2a). Finally, the complementation cassette was ligated upstream to the miniCR cassette into the respective pENTRY vectors (Quick-Ligase, NEB), generating ENTRY-SB2-CC, pENTRY-SB3×6-CC, and pENTRY-Ctrl-CC, respectively which were recovered by *E. coli* propagation (One Shot TOP 10, ThermoFisher Scientific).

**Culturing and transfection.** *S. solfataricus* M18, an uracil–auxotrophic derivate of P1 (DSM 1616, ATCC 35091), was cultivated under shaking conditions at 78 °C, pH 3, in Brock T/S/U medium supplemented with 0.2% (+) D-Sucrose (S) (Serva) (wt/vol), 0.1% tryptone (T) (Roth) (wt/vol) and uracil (U) (Sigma Aldrich) at a final concentration of 0.0125 mg per ml[35]. For transfection, 50 μl of competent cells[51] were separately mixed with 150 ng of pDEST-SB/Ctrl vector DNA, respectively, and electroporated using 1-mm cuvettes in a Gene Pulser Xcell (Biorad) under the following conditions: 1250 V, 1000 Ω, 25 μF. After 1 h recovery, an inverse plaque assay was performed by pouring a 0.4% gellan gum solution (GELRITE, Roth) containing 2 μl of the transfection solution and 10× concentrated M18 cells on uracil-free Brock-medium plates[35]. After 2 days of incubation, three plaques per transfectant were each isolated, transferred to liquid Brock T/S medium and incubated under the above-described conditions. Transformation of pIZ-plasmids was performed analogously with modified electroporation—settings: 1240 V, 25 μF, 1000 Ω. After recovery, the transformation solution was directly transferred to Brock NZ/S liquid medium [0.1% N-Z-amine AS (NZ) (Fluka Analytics) (wt/vol), 0.2% D-sucrose (Serva) (wt/vol)] referred to as "primary cultures". Single colony purifications were performed by overlay plating, where 12 μl of transformation solutions were mixed with 10 ml of Brock NZ/S medium supplied with 0.4% gellan gum (GELRITE, Roth) and poured on uracil-free NZ/S plates. Single colonies were isolated after 10 days and were further transferred to Brock NZ/S liquid medium. For incubation of cells transformed with complementation constructs (CC), D-arabinose (*BioChemica*, AppliChem) instead of sucrose was supplied to growth media to same concentrations.

**Growth assays, sampling, and culture PCR.** Growth of control and silenced plaque cultures was monitored at OD (600 nm) (Beckam Coulter, DU 800 Spectrophotometer), and samples were taken daily to verify the integrity of the miniCRs by culture PCR. For this, 2 μl of the culture were used in a PCR reaction (Phusion Polymerase, ThermoFisher, Scientific) together with 406-FW and 406-RV primers binding the vector backbones ~ 100 bp up- and downstream of the miniCR array (Supplementary Table 1). Entire reaction volumes of each culture PCR sample were loaded on 1% agarose gels together with an 1 kb Plus DNA ladder (Thermo Scientific) to determine fragment length. Uncropped gel pictures are provided in the Source Data[70]. In total, 5 ml of $OD_{600} = 0.1$ cultures were harvested cultures for RNA-, DNA-, and S-layer preparations, respectively.

**DNA, RNA, and cDNA preparation.** DNA and RNA were isolated as described in ref. [34] followed by further RNAse (Omega, bio-tek) and DNAse (TURBO DNA-free kit, ThermoFisher Scientific) treatment of extracted DNA and RNA, respectively, following the manufacturer's protocol. The integrity of extracted RNA was verified by gel electrophoresis and DNA digestion was checked by PCR. One microgram of RNA was reverse transcribed using ProtoScript II Reverse transcriptase (NEB) according to the manual and further purified (Monarch, PCR & DNA Cleanup Kit, NEB). Nucleic acid concentrations were determined fluorometrically by Qubit using RNA BR-, DNA BR-, and ssDNA Assay Kit for cDNA, respectively (ThermoFisher Scientific).

**Quantitative RT-PCR analysis.** All quantitative PCRs were performed using GoTaq qPCR Master Mix (Promega) in an Eppendorf Mastercycler ep*gradient* S relplex2 (Eppendorf). Three technical replicates of each biological replicate of every experiment were measured, respectively. Transcripts of *slaB* and the SSOP1_3283 housekeeping gene (glyceraldehyde-3-phosphate dehydrogenase), as well as 16S rRNA were quantified from same cDNA preparations using primer pairs qSlaBFW/RV, q3194FW/RV, and q16SFW/RV, respectively (Supplementary Table 1). Relative expression of the *slaB* gene was assessed by calculating the $\Delta C_t$ of the amplicons of the two primer pairs and normalizing the values to Ctrl control cultures[34]. For infection assays (see below), virion numbers were quantified directly on 5 μl of the solution using primers qD291FW/RV (binding ORF A291 of SSV1, see Supplementary Table 1). Standards used in qPCRs were linearized pDEST-MJ vectors (*Xho*I, ThermoFisher, Scientific) containing the appropriate gene. qPCR efficiency was between 100 and 84% in all runs.

**Light microscopy.** For light and fluorescence microscopy, 5 μl of fresh cultures were transferred to a microscope slides (Marienfeld superior) and immediately imaged using a Jenoptik microscope camera (ProgRes MF Cool CCD 1.4 M.P) attached to a Nikon ECLIPSE Ni-E. For viability staining, 1 ml of freshly harvested

cells were incubated with Syto9 and PI (Invitrogen LIVE/DEAD BacLight Bacterial Viability Kit) according to the manufacturers protocol and imaged (PI: Nikon TRITC Bandpass filter cube; Syto9: Nikon FITC Bandpass filter cube). For Hoechst (Invitrogen H1399) staining, cells were fixed at 4 °C overnight in 70% ice cold ethanol, subsequently washed and resuspended in buffer A (20 mM TRIS-Acetate buffer pH 5.5), incubated according to the manufacturers protocol and imaged.

**Electron microscopy.** Electron micrographs of negatively stained cells were produced from freshly harvested cells ($OD_{600} = 0.1$), which were washed once with buffer A and resuspended in 100 μl of buffer A. In total, 30 μl of the cell suspension were absorbed onto carbon-coated copper grids and negatively stained with 2% uranyl acetate. For thin sections, 2.5% glutaraldehyde was added to 2 ml of cells harvested at OD 0.1 and incubated for 2 h. After which the cells were further fixed with 0.1% osmium tetroxide in 0.02 mM sodium cacodylate overnight. The pellet of fixed cells was embedded into Low Viscosity Resin (Agar scientific; Agar Low Viscosity Resin Kit) and then cut into 60 nm thin section (LEICA EM UC6). The sections were absorbed onto formvar coated copper grids and contrasted (2.,5% Gadolinium acetate, 30 min; 3% Lead citrate, 8 min). Both negatively stained samples and thin sections were analyzed using a Libra 120 (Zeiss). For scanning electron microscopy, late exponential cells were fixed (2.5% glutaraldehyde), washed (0.02 mM sodium cacodylate), spotted onto 0.01%-poly-L-lysine-coated glass slides and dehydrated using a graduated ethanol series followed by critical point drying (Leica EM CPD300). The slides were subsequently placed on conductive stubs, gold coated for 60 s (JEOL JFC-2300HR) and analyzed (JEOL JSM-IT300).

**Flow cytometry.** Cell size (FSC), cell granularity (SSC), and glycoprotein (FITC) analysis was performed on a BD FACSCantoII. For FSC and SSC measurements, 50 μl of a growing culture were diluted 1:10 in buffer A. For glycoprotein quantification, 20 μl of freshly harvested cells were diluted 1:20 in buffer A and subsequently incubated with 40 μg per ml Wheat Germ Agglutinin, Alexa Fluor 488 Conjugate (W11261, ThermoFisher Scientific) for 30 min in the dark at room temperature and analyzed. To measure the DNA content, cells were fixed at 4 °C overnight in 70% ice-cold ethanol, subsequently washed and resuspended in buffer A and stained with Hoechst 33342 (Invitrogen H1399) according to the manufacturers protocol. In all, 50 μl of the stained cells were diluted 1:10 in the washing buffer and analyzed (MoFlo Astrios. Beckman Coulter).

For cell counting, 20 μl of freshly harvested cells were diluted 1:20 in buffer A and cells per ml were calculated from flow rate (10 μl per min), events recorded per second and recording time (see Source Data file[70]). This method was verified by comparison with cell counts by light microscopy.

**NaCl assay.** Aliquots of growing SB3×6 and Ctrl plasmid transformants were adjusted to same cell concentrations (cell number assessed by microscopy) and incubated under normal growth conditions for 2 h (78 °C, shaking). NaCl (Roth) was added to the shaking cultures to a final conc. of 100 mM followed by 45 min incubation. Consecutively, concentration was increased to 200 mM followed by 45 min incubation. 5 μl of cultures were sampled in triplicates right before NaCl addition and after each NaCl incubation period respectively, without interrupting the incubation process. Samples were applied on microscope slides and immediately imaged by phase contrast microscopy (see above). Diameter of cells ($n = 50$ per condition and culture) were measured at spots of widest distance on micrographs taken of each culture using MicrobeJ[71].

**S-layer composition analysis.** S-layer was purified as previously described[38] with small alterations, in short: 30 ml of a growing culture at $OD_{600} = 0.1$ were harvested by centrifugations at $2500 \times g$ and 4 °C for 15 min. The subsequent pellet was resuspended in 2 ml of buffer 1 (10 mM NaCl, 1 mM PMSF, 20 mM MgSO₄, 0.5% sodium lauroylsarcosin, pH  6) and 10 μl DNAse (TURBO DNA-free kit, ThermoFisher Scientific) was added. Subsequently the solution was incubated at 45 °C for 30 min. Following centrifugation at $13,000 \times g$ for 15 min, the supernatant was discarded, and the procedure repeated. After the second round of centrifugation, the pellet appeared to have two distinct fractions. The top white fraction was carefully resuspended in 2 ml of buffer 2 (10 mM NaCl, 20 mM MgSO₄, 0.5% [wt/vol] SDS, pH  8), incubated at 45 °C for 30 min and centrifuged at $13,000 \times g$ for 15 min. The incubation with buffer 2 was repeated, until the pellet was completely translucent, at which point it was resuspended in 50 μl of MQ. Purified S-layer samples were denatured and run on a precast gel (NuPAGE 4–12% Bis-Tris Protein Gels). The protein band observed at ~ 40 kDa was excised (UniProt Acession: A0A0E3K5T7—slaB = 41 kDa), processed for mass spectrometric analysis and subsequent protein identification as described in ref. [72].

**Denaturing and reassembly of SlaA.** S-layer was purified as described above and washed another five times in buffer 2, in order to remove SlaB from the S-layer preparation[38]. The resultant S-layer extract was pelleted by centrifugation ($13,000 \times g$ for 15 min), resuspended in 20 mM Na₂CO₃ (pH 10.5) as described in ref. [45]. The resuspended S-layer was incubated under agitation at room temperature overnight, after which the S-layer sacculi were no longer visible using Light Microscopy. In total, 10 × concentrated Brock medium was added to the denatured

proteins to a final concentration of 1×, after which the pH was titrated to pH 3.5 by the addition of $H_2SO_4$. The resulting solution was incubated overnight at 78 °C. After incubation, aggregates were observable.

**Infection assay.** Virion solutions for infection assays were prepared by sterile filtering (filter: 200 nm pore size) of supernatants of a SSV1—producing *S. solfataricus* culture. Virion solutions were further verified to be pure of cell contaminants by microscopy (see above). Virus copy numbers in the virion solution were quantified by RT-qPCR using qD291FW/RV primers (see Quantitative RT-PCR analysis and Supplementary Table 1). Fresh SB3×6 and Ctrl–plasmid transformants were harvested at $OD_{600} = 0.1$ and adjusted to same cell numbers of ~5 × $10^7$ cells per ml (cell numbers assessed by FC, see Flow Cytometry) by transferring aliquots to pre-warmed medium, creating three replicate cultures. Each replicate was independently mixed with SSV1 virions at a distinct MOI (0.1, 5, or > 100). As control, the same amount of the virus was added to cell-free medium and treated under same conditions. All cultures were incubated under standard conditions (78 °C, shaking) and 1 ml samples were taken at regular intervals, without interrupting the incubation process. Each sample was centrifuged for 10 min at 11,000 × g followed by separation of the supernatant from the cell pellet and further filtering of the supernatant (450 nm pore size). Cell pellets were washed with fresh Brock NZ/S medium (see Culturing) three times (centrifugation at 11,000 × g) to remove residual virus in the supernatant and finally resuspended in Brock NZ/S medium. In total, 5 μl of filtered supernatant and the cell pellet solution respectively, were directly used as templates in separate RT-qPCR measurements (three technical replicates performed per biological replicate) each using qD291FW/RV primers (see Quantitative RT-PCR analysis and Supplementary table 1). Virus copy numbers measured in supernatants and resuspended cell pellets were normalized to those of initial virus added (as measured in the cell-free control) and to the cell count, which was assessed by FC (see Flow Cytometry) for each individual sample. For analysis, change of virion number was determined in comparison with time-point 10 min PI.

**Virus assays.** For SSV1 infection assays at MOI 5, plaque assays were performed to assess the number of extracellular virions in supernatants sampled at different timepoints from infected SB3×6/Ctrl plasmid transformants and cell-free controls (see Infection assay), respectively. Different dilutions of every supernatant sample were each mixed with 10 × *S. solfataricus* P1 in a 0.4% gellan gum solution and poured on Brock-medium plates (see Culturing). Plates were incubated for 48 h at 78 °C. The number of plaques obtained from SB3×6 and Ctrl supernatants was compared with the number of plaques present in a cell-free virus control. Spotting assays were performed by using different dilutions of SSV1 virion solutions (purified as described in "infection assay") and spotting them on lawns of 10 × concentrated cells of SB3×6 and Ctrl plasmid cultures. Biological replicates represent independently incubated SB3×6 and Ctrl cultures.

**Reporting summary.** Further information on research design is available in the Nature Research Reporting Summary linked to this article.

## Data availability
The source data underlying Figs. 1b–f, 2c–e, 4, 5b–d, 6a, c and 7b, c, Table 1, and Supplementary Figs. 3a, b, 4, 7, 11, and 12a, b are provided as a Source Data file and are available at figshare[70].

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

## Acknowledgements

We thank Stephan Köstlbacher for help with data imaging, as well as Melina Kerou and Filipa Sousa for technical advice and scientific discussions. EM work was performed at the Core Facility Cell Imaging and Ultrastructure Research, University of Vienna–member of the Vienna Life-Science Instruments (VLSI) with special thanks to Norbert Cyran, Daniela Gruber and Siegfried Reipert. Also, special thanks go to all members of the master course "MMEI-(300 307)" 2017, who provided first results on the silencing experiment. This work was financed through a DOC-fellowship of the Austrian Academy of Sciences (OEAW) and a PhD completion grant of the University of Vienna to IZ, through P 29399-B22 of the Austrian Science Fund (FWF) to BS, as well as ERC-Adv grant TACKLE (No. 695192) to CS.

## Author contributions

I.Z., K.P., and E.W. performed experiments, C.S. and I.Z. conceived this study, I.Z. and K.P. wrote first manuscript draft, C.S., I.Z., K.P., B.S., E.W., and U.S. discussed results and manuscript.

## Competing interests

The authors declare no competing interests.
