## [Peer Review File · Nature Communications]

Reviewers' comments:

Reviewer #1 (Remarks to the Author):

In this manuscript Zink et al. report the roles of the S-layer in the archaeon *Sulfolobus solfataricus*. The authors used the endogenous type III CRISPR-Cas system to silence the S-layer *slaB* gene expression. They conclude that the type III CRISPR system is an effective tool for gene silencing and the S-layer is vital for cell division and virus infection.

The paper was very well written and the data were clearly presented. However, the novelty of the work remains to be considered carefully. First, gene silencing using an endogenous type III CRISPR-Cas has already been described several times in the literature, e.g. references 30 – 33 cited in the manuscript. Second, a recent paper published in *Nat. Commun.* demonstrated the non-essentiality of *SlaB* as well as the partner protein *SlaA* in a very closely related *Sulfolobus* species, *S. islandicus* (reference 64 in the paper). Knocking out both *slaA* and *slaB* genes was achieved in *S. islandicus*. The contradictory results might be due to, as the authors stated in Discussion, the different cell cultivation conditions, namely with or without agitation. Therefore, the vital role of the S-layer in cell division, which was claimed in this paper, appears to be conditional.

Major comments:

1. One of the major claims of the paper is the employment of type III CRISPR for gene silencing. However, knockdown clones (colonies) could not be obtained when the plasmid-based SB constructs were used. Instead, the authors used the "primary cultures", mixtures of both transformed and non-transformed cells. Although subsequent cultivation in selection medium could eliminate the non-transformed cells, this method may suffer complications in data interpretation. Would it be possible to adjust the spacer sequences and/or lengths to reduce RNA cleavage by type III CRISPR-Cas, so that colonies could be obtained? Subsequent induction of the crRNA would then allow phenotype analyses.
2. Another major claim of the paper is the role of S-layer in virus infection. Ambiguities in this part include the following.
 - A. Fig. 7b shows less SSV1 DNA in the knockdown "primary cultures" 1 hour after infection, and the authors suggest that *SlaA* or *SlaB* is SSV1 attachment site (lines 349-357). Given the possibility that SSV1 already started replication at 1 h post infection, the DNA level difference is difficult to be interpreted. Measuring extracellular SSV1 DNA at different times post infection would be a better approach to determine the attachment role of *SlaA* and *SlaB*.
 - B. in Fig. 7c, virus release was compared between pDEST-SB2 and pDEST-MoE transfectants. This comparison is inappropriate, as pDEST-SB2 transfectant grows much slower than pDEST-MoE transfectant (Fig. 1b). As shown in Fig. 1b, it takes about 65 hours for pDEST-MoE transfectant, but 105 hours for pDEST-SB2 transfectant, to reach OD600 0.1. As such, it is not clear whether the difference in SSV1 particle numbers was due to the difference in incubation time or in release efficiency.

Minor comments:

1. Fig. 1C, no growth curve for cells containing SB3?
2. Lines 167-168, "Growth profiles of the complemented cultures (SB3x6-CC) were almost identical to control MoE", this is not true. The complementation indeed rescued the growth phenotype to some extent, but significant growth retardation is still obvious in the complemented strain in comparison to control MoE.
3. Fig. 4, why is there no data for SB2-CC?
4. Fig. 5, the total fraction for MoE is clearly more than 100% (43% of 1.5 micrometer, 65% of 2 micrometer). Is this a mistake?
5. Lines 375-378, I am not convinced that the cultivation condition (with agitation) of this work resembles better the natural habitats of *Sulfolobus*, which are abundant in hot spring sediments as

well.

6. Line 522, " 11 0000 g", should it be 11 000 g?

Reviewer #2 (Remarks to the Author):

In this submission, Zink and co-authors consider the roles of the archaeal S-layer in cell division and viral infection upon CRISPR-mediated silencing of *slab*, encoding one of the two S-layer glycoproteins in *Sulfolobus solfataricus*. The introduction of this gene silencing strategy into investigations of the S-layer is welcome, given challenges in deleting S-layer-related genes using other approaches. Still, there are numerous problems that must be addressed, many related to the conclusions drawn. Specifically, many of the observations reported could be due to reduced S-layer glycosylation (shown by several labs to be important for numerous functions in archaea, yet largely overlooked here) and not reduced S-layer coverage. The authors need to discern between the two scenarios before they can conclude much of what they conclude.

My comments are listed in the order the points addressed were encountered while reading the text.

Line 28 – The claim that 'experimental analysis of its physiological impact on the living cell has so far been outstanding' is misleading. Work from the Eichler, Albers and Jarrell labs has addressed the importance of S-layer glycoprotein glycosylation in living cells. Pohlschroder has considered S-layer glycoprotein membrane-anchoring in living cells. Indeed, many of these references are cited in the subsequent text. Moreover, the phrase 'has so far been outstanding' can be replaced with something better.

Line 36 – Likewise, evidence for the importance of S-layer glycoprotein glycosylation for virus-host interactions has been shown by Roine, Gophna and Eichler, which are also cited in the text. Again, a misleading claim.

Line 56 – This is not correct. In halophilic archaea, the S-layer is anchored to the cell surface via covalently-linked lipid anchors.

Line 58 – Gram+ and Gram-

Line 75: Please cite studies showing this point in Archaea

General comment: The title of the manuscript refers to archaeal S-layers yet the Introduction does a poor job of discerning the bacterial and archaeal versions of this structure. It is not clear why the authors include a reference to a bacterial S-layer in every example listed.

Line 77 – Archaeal S-layer glycosylation has been shown to be important for many more cell functions. See works by Albers, Jarrell and Eichler, for example.

No mention of S-layer removal being a first step in numerous strategies used to transform a variety of archaeal strains.

Line 81: Misleading. A role for S-layer glycosylation was reported in viral infection of a halophilic archaea by Eichler and Roine (ref 63 in this manuscript).

Line 188: Can the authors provide statistics on how many cells they observed lacking gaps in the S-layer and the sizes of these gaps? While the figures shown are impressive, they are single examples.

Line 199: Has S-layer morphogenesis been studied in Archaea?

General comment: Why not assign the different strains names that mean something to the reader? MoE-CC – surely, there is a friendlier way to name this strain.

Line 202: It is not clear what the authors are seeing. There is no control for the specificity of lectin (not identified here, only in the fig legend) binding (e.g. upon competition with added sugar, after chemical deglycosylation). Has it been shown that this particular ligand binds the glycan attached to *S. solfataricus* proteins? Glycolipids? The equivalent of a western blot using the lectins would be welcome. Does the decrease in glycosylation correspond with the extent of SlaB deletion? Why would introduced *S. islandicus* SlaB be differently glycosylated in *S. solfataricus* than would the native SlaB? Host-dependent glycosylation in Archaea has been demonstrated.

Line 211: The glycosylated nature of SlaB was previously reported (ref 51 in this manuscript).

General question: Despite a reduced level of SlaB, is there no compensating self-assembly of SlaA? The authors mention the possibility of non-membrane-anchored SlaA-based constructs in the medium later in the Discussion but provide no evidence.

General comment: Albers has shown that in *S. acidocaldarius*, protein glycosylation is essential for survival. It was proposed that the water-trapping sugar coating on the cell surface offers an advantage in a thermoacidic environment. Could the effects seen in this manuscript not be due to decreased surface glycosylation resulting from the perturbed S-layer rather than the perturbed S-layer itself? This could be tested by using the CRISPR technology employed here on enzymes of glycosylation (e.g. AgIB).

Line 234: Aren't these finding opposite of those reported by Albers (Meyer et al 2013)?

General question: Are the SlaB-lacking cells motile? Assessing this would provide novel information on the importance of the S-layer for archeallum construction, stability and function. Such studies would help distinguish the importance of the S-layer as opposed to S-layer glycosylation.

Line 376: Any support that the strain actually lives in the bubbling hot spring and not where no bubbling? No clusters or biofilms in nature?

Reviewer #1 (Remarks to the Author):

In this manuscript Zink et al. report the roles of the S-layer in the archaeon *Sulfolobus solfataricus*. The authors used the endogenous type III CRISPR-Cas system to silence the S-layer *slaB* gene expression. They conclude that the type III CRISPR system is an effective tool for gene silencing and the S-layer is vital for cell division and virus infection.

The paper was very well written and the data were clearly presented. However, the novelty of the work remains to be considered carefully. First, gene silencing using an endogenous type III CRISPR-Cas has already been described several times in the literature, e.g. references 30 – 33 cited in the manuscript. Second, a recent paper published in *Nat. Commun.* demonstrated the non-essentiality of *SlaB* as well as the partner protein *SlaA* in a very closely related *Sulfolobus* species, *S. islandicus* (reference 64 in the paper). Knocking out both *slaA* and *slaB* genes was achieved in *S. islandicus*. The contradictory results might be due to, as the authors stated in Discussion, the different cell cultivation conditions, namely with or without agitation. Therefore, the vital role of the S-layer in cell division, which was claimed in this paper, appears to be conditional.

We thank the reviewer for the comment. We will first discuss the novelty of the silencing approach used and then elaborate on the phenotypical differences observed in our study, in comparison to the mentioned reference.

Previously published studies, by our group and others, applied the type III system exclusively to silence reporter genes as proof of principle (Zebec *et al.*, 2014, 2016, Peng *et al.*, 2014), or for targeting a non-essential chromosomal gene not triggering any phenotypic effects (e.g. no growth retardation) (Han *et al.*, 2017). One novelty of our study lies in the chosen target gene. We show for the first time, that the established CRISPR type-III system can be adapted and extended to study the function of putatively essential genes, which are difficult or impossible to be deleted. We introduce new vectors (virus and plasmid-based), which can be readily used by researchers in the field for conducting such silencing experiments. Furthermore, this study is the first to show that the intensity of the silencing phenotype can be controlled by the stepwise reduction of gene transcripts, thus allowing researchers to elucidate the function of an essential gene in a progressive manner (cf. SB3, SB3x2, SB3x6 Fig. 1, new Fig. 4). Moreover, we expanded this tool by expressing not only a silencing miniCRISPR but also a complementing gene from the same vector (complementation constructs). Such a construct paves the way to easily investigate functions of a protein that potentially complements the function of the silenced gene.

The diverging results of our study compared to the study in *S. islandicus* might be due to the fundamentally different experimental approaches applied. Full gene deletions, such as applied to *S. islandicus*, rely on selection processes, which are prone to trigger genetic compensation reactions. Such reactions as a response to a gene disruption are common, as they were only recently reported to be a problem in eukaryotic genetics (El-Brolosy, *et al.* 2019). The depleted *SlaB* could potentially be rescued by the overexpression of another membrane protein of as yet unknown function. Posttranscriptional silencing, as applied in our study, does not alter the DNA and therefore potentially minimizes the risk of triggering genetic compensation reactions. This is also reflected by the fact, that cultures rather alter or lose the silencing

miniCRISPR, than mutating the targeted gene. We therefore believe, that a silencing approach might better illustrate the role of the gene, which we do believe is an essential one. We have reformulated the discussion part for this section to be clearer in the manuscript.

Major comments:

1. One of the major claims of the paper is the employment of type III CRISPR for gene silencing. However, knockdown clones (colonies) could not be obtained when the plasmid-based SB constructs were used. Instead, the authors used the "primary cultures", mixtures of both transformed and non-transformed cells. Although subsequent cultivation in selection medium could eliminate the non-transformed cells, this method may suffer complications in data interpretation. Would it be possible to adjust the spacer sequences and/or lengths to reduce RNA cleavage by type III CRISPR-Cas, so that colonies could be obtained? Subsequent induction of the crRNA would then allow phenotype analyses.

We thank the reviewer for the comment. We believe that the use of primary cultures is not an issue of concern, as phenotypes were always compared to control cultures (= Ctrl) which have been cultured and transformed the same way (Ctrl and all others, -CC are primary cultures). Furthermore, the same phenotypes were observed when using the complementary approach employing a virus vector, where cells grown from single-plaque isolates were investigated (Fig. 1b, d, Supplementary Fig. 3,4). In general, we think that the use of primary cultures might even be a welcoming approach, as it makes "real-time monitoring" of the developing silencing phenotype possible, whereas selection by plating can trigger compensatory effects (as elaborated in the previous comment and in the discussion part). The reviewer's suggestion to adjust the spacer sequence/lengths to reduce RNA cleavage is indeed very interesting and we agree that this would be a complementing approach. As such an approach would demand a whole new set-up of the experiment, and establishment of an inducible promoter system (as most promoters used in *S. sofataricus* are very leaky and would potentially trigger silencing already in an uninduced state), we are of the opinion that this is not in the scope of the study.

2. Another major claim of the paper is the role of S-layer in virus infection. Ambiguities in this part include the following.

A. Fig. 7b shows less SSV1 DNA in the knockdown "primary cultures" 1 hour after infection, and the authors suggest that SlaA or SlaB is SSV1 attachment site (lines 349-357). Given the possibility that SSV1 already started replication at 1 h post infection, the DNA level difference is difficult to be interpreted. Measuring extracellular SSV1 DNA at different times post infection would be a better approach to determine the attachment role of SlaA and SlaB.

We thank the reviewer for this constructive remark and we agree that this is a justified criticism. We therefore performed new experiments, according to the reviewer's suggestion, which were embedded in the results part of the paper (see new Fig. 7b, new Table 1), starting with: '*To assess the efficiency of virus uptake over time....*'

We have conducted three additional infection assays (three biol. replicates each) at different MOIs (0.1, 5, >100). For each, extracellular SSV1 DNA was measured at different time points post infection (10 min PI, 30 min PI and 60 min PI) and compared to the initial virus added (which was also added to a cell-free control and measured). These measurements were complemented by quantification of intracellular virus copies

per cell, as performed in experiments for the initial submission of this manuscript. The data collected in these new experiments confirmed the conclusions drawn from the initially submitted data: A significantly more efficient uptake of virions in control cultures compared to the S-layer depleted cultures. We observed that significantly less extracellular virus DNA was left in supernatants of Ctrl cultures already after 30 min PI than in SB3x6 silenced cultures. Simultaneously an increase of intracellular viral-DNA was detectable at respective time points. This indicates, as initially discussed, a less efficient SSV1-uptake and therefore lower SSV1 susceptibility of S-layer depleted cells. Most strikingly, the trend was independent of the MOI (as listed in the new Table 1). All measurements (as in the initially submitted manuscript) were conducted by qPCR. Additionally, to prove the reproducibility of the results using a different experimental approach, for samples at MOI 5, a plaque assay was performed using the filtered supernatant of the different timepoints on wild type P1 cell lawns. When comparing timepoints T30 and T50 PI, fewer plaques (compared to a control plate of the cell-free virus control) were obtained on plates with Ctrl culture filtrate.

B. in Fig. 7c, virus release was compared between pDEST-SB2 and pDEST-MoE transfectants. This comparison is inappropriate, as pDEST-SB2 transfectant grows much slower than pDEST-MoE transfectant (Fig. 1b). As shown in Fig. 1b, it takes about 65 hours for pDEST-MoE transfectant, but 105 hours for pDEST-SB2 transfectant, to reach OD600 0.1. As such, it is not clear whether the difference in SSV1 particle numbers was due to the difference in incubation time or in release efficiency.

We agree that incubation time could also be the cause of this effect. We don't see how we could distinguish both possibilities at the moment, and since this was not a main result of the paper, we have taken it out of the manuscript.

Minor comments:

1. Fig. 1C, no growth curve for cells containing SB3?

Thank you for the note. We have added SB3 to the growth curve in new Fig. 1c.

2. Lines 167-168, "Growth profiles of the complemented cultures (SB3x6-CC) were almost identical to control MoE", this is not true. The complementation indeed rescued the growth phenotype to some extent, but significant growth retardation is still obvious in the complemented strain in comparison to control MoE.

We agree. We have re-formulated this section, starting: '*Growth of complemented cultures (SB3x6-CC) has significantly recovered in comparison to the strong growth retardation observed for the non-complemented SB3x6 transformants cultivated in parallel (Fig. 2c, Supplementary Fig. 4)*'.

We believe that the observable difference between the control (name changed from MoE to Ctrl in response to reviewer 2) and SB3x6-CC growth curve (which is not significant until timepoint "312 hours") resulted from CRISPR-type III cross-targeting of the complemented *slaB*-M.16.4 mRNA. The targeting crRNA-SB3 and crRNA-SB2 mismatched the complementary protospacer region on the *slaB*-M.16.4 mRNA in nine and four positions, respectively (cf. Fig. 2B). This number of mismatches might still allow binding and degradation of the complementing mRNA to a certain degree, as it was shown that all CRISPR systems can tolerate mismatches (e.g. Manica *et al.*, 2014,

NAR). However, in all experiments the silenced cultures were considerably more affected than CC cultures indicating convalence of silencing.

Another explanation could be that the expression of the complementing SlaB from the vector might be lower.

3. Fig. 4, why is there no data for SB2-CC?

The glycosylation level was re-examined with fresh cultures of all transformants, including SB2-CC (also in response to reviewer 2, comment 14). A new graph was therefore produced (new Fig. 4), including SB2-CC which showed glycosylation levels comparable to the control culture.

4. Fig. 5, the total fraction for MoE is clearly more than 100% (43% of 1.5 micrometer, 65% of 2 micrometer). Is this a mistake?

We apologize for this obvious mistake. The wrong excel column was selected in formula for the calculation of MOE-control data. Corrected Data still shows the same general result.

5. Lines 375-378, I am not convinced that the cultivation condition (with agitation) of this work resembles better the natural habitats of *Sulfolobus*, which are abundant in hot spring sediments as well.

We agree that *Sulfolobus solfataricus* might also thrive in sediments or without bubbling, but all dozens of strains isolated by Zillig and collaborators (including one of us) were isolated from the inner, most strongly bubbling zone of hot springs (for example those described in Zillig et al. 1994, Systematic and Applied Microbiology, 16, 609-628) but also *Sulfolobus solfataricus* P1 used in this study. In contrast, *S. acidocaldarius* was probably isolated from the edge of hot springs (by Thomas Brock in the 1980s). To our experience, shaking conditions result in higher growth rates, indicating that *Sulfolobus* is well adapted to it.

6. Line 522, " 11 0000 g", should it be 11 000 g?

We thank the reviewer for pointing out this mistake and apologize for the typo.

Reviewer #2 (Remarks to the Author):

In this submission, Zink and co-authors consider the roles of the archaeal S-layer in cell division and viral infection upon CRISPR-mediated silencing of slab, encoding one of the two S-layer glycoproteins in *Sulfolobus solfataricus*. The introduction of this gene silencing strategy into investigations of the S-layer is welcome, given challenges in deleting S-layer-related genes using other approaches. Still, there are numerous problems that must be addressed, many related to the conclusions drawn. Specifically, many of the observations reported could be due to reduced S-layer glycosylation (shown by several labs to be important for numerous functions in archaea, yet largely overlooked here) and not reduced S-layer coverage. The authors need to discern between the two scenarios before they can conclude much of what they conclude.

We thank the reviewer for this critical comment and agree that glycosylation should be better introduced and should be included when discussing our results. We have therefore extensively included earlier studies on the function of glycosylation in archaea in the introduction and also addressed this aspect in the discussion. In our study, the native S-layer, i.e. the glycosylated protein is in the center of our study and we have considered it as a unit. We have not attempted to discern which parts of the S-layer cause the phenotypic effects, but we agree that in the light of earlier work on glycosylation, it makes sense to discuss this aspect. We have made this more clear in the text and also discuss our findings accordingly now (see e.g. below No. 2).

My comments are listed in the order the points addressed were encountered while reading the text.

1) Line 28 – The claim that ‘experimental analysis of its physiological impact on the living cell has so far been outstanding’ is misleading. Work from the Eichler, Albers and Jarrell labs has addressed the importance of S-layer glycoprotein glycosylation in living cells. Pohlschroder has considered S-layer glycoprotein membrane-anchoring in living cells. Indeed, many of these references are cited in the subsequent text. Moreover, the phrase ‘has so far been outstanding’ can be replaced with something better.

We agree with the reviewer that the statement did not take into account the work on glycosylation and have reformulated the abstract to:

‘In most Archaea, a glycosylated S-layer constitutes the sole cell wall and there is evidence that it contributes to cell shape maintenance and stress resilience. However, to resolve its function, comprehensive in vivo studies using targeted S-layer removal are required’

2) Line 36 – Likewise, evidence for the importance of S-layer glycoprotein glycosylation for virus-host interactions has been shown by Roine, Gophna and Eichler, which are also cited in the text. Again, a misleading claim.

We have refined the formulation at the end of the abstract to:

‘This study demonstrates a CRISPR type III system as an effective tool for essential gene silencing and provides experimental evidence that an intact S-layer plays vital roles in virus susceptibility and cell division..’

We here want to explain why we think that our study differs from the ones mentioned by the reviewer and why we are convinced that ours is the first one experimentally showing

the indispensability and therefore essentiality of an intact S-layer glycoprotein for a virus infection.

We do agree, that there are indications for the importance of the S-layer in virus infection from the above- mentioned authors, and we have additionally referenced some in the respective section in the introduction (see comment 9 of reviewer 2).

However, to our knowledge, there is only one study experimentally addressing the role of S-layer in virus infection (Kandiba, 2012, ref. 32 in new manuscript and by extension the follow – up study by Zaretsky *et al.*, 2018), which in our opinion does not explicitly distinguish between glycosylated S-layer and other glycosylated surface structures.

Kandiba *et al.*, 2012 showed reduced infection of the host *Halorubrum* sp. PV6 by HRPV- 1 virus performing an “infection inhibition assay” at a single MOI 0.0000025 with 3 replicates. The inhibition assay relied on the addition of a sialic acid which masked the putative binding site of the virus, a 5-N-formyl-legionaminic, the last sugar of the N-linked glycan verified on the S-layer protein. A reduction of infection was only shown after 4 hours pi. However, given the possibility, that the same N-glycan modification, or at least, the 5-N-formyl-legionaminic, is shared by additional surface proteins (glycosylated surface proteins are abundant as listed in Jarell and Adams, 2005), which seems common in Archaea (e.g. flagella and S-layer proteins, cytochrome and S-layer protein share a glycan sequence as shown in Tripepi *et al.*, 2012, Voisin *et al.*, 2005, Peyfoon *et al.*, 2010), Kandiba and co-authors do not explicitly show that that the HRPV-1 infection is dependent on the presence of the S-layer glycoprotein.

Contrasting to Kandiba *et al.*, 2012, we clearly show that infection is dependent on the presence of the S-layer, as its targeted depletion efficiently reduced virus uptake. In our study, we cannot distinguish which structure on the S-layer is responsible for this effect, the glycosylation might indeed be involved (as we discussed in the paper also citing Kandiba *et al.*), but as stated above we consider the glycosylation to be a part of the S-layer protein. We have made this clearer in the text now.

During this round of revision (and in response to reviewer 1), we have performed additional experiments to strengthen our results. We demonstrate reduced virus uptake at different times post infection. Significant differences between control and silenced cells were already obvious after 30 min p.i. which was verified in 10 biological replicates at three different MOIs (see new Fig. 7b and new Table 1).

3) Line 56 – This is not correct. In halophilic archaea, the S-layer is anchored to the cell surface via covalently-linked lipid anchors.

We thank the reviewer for this important amendment and corrected the statement to:

“This is a major difference to most Archaea, where the S-layer serves as the sole cell wall component, which is in direct contact to the cytoplasmic membrane. Binding is either achieved by a hydrophobic anchoring as in the hyperthermophile *Sulfolobus*, or via a covalent lipid – linkage which seems predominant in halophilic Archaea.”

4) Line 58 – Gram+ and Gram-

We thank the reviewer for pointing this out and changed it accordingly in the manuscript.

5) Line 75: Please cite studies showing this point in Archaea

We did not find an equivalent study in Archaea and deleted the sentence (see comment below).

6) General comment: The title of the manuscript refers to archaeal S-layers yet the Introduction does a poor job of discerning the bacterial and archaeal versions of this structure. It is not clear why the authors include a reference to a bacterial S-layer in every example listed.

We appreciate the constructive criticism and we revised the introduction of the manuscript in sections mentioned below. We have focussed more on the Archaeal part now by adding additional archaeal work and relevant literature, especially also focussing on glycosylation and deleting some of the bacterial references.

7) Line 77 – Archaeal S-layer glycosylation has been shown to be important for many more cell functions. See works by Albers, Jarrell and Eichler, for example.

We have considerably modified this section of the introduction by referring to glycosylation of the archaeal S-layer and including studies of phenotypic effects of glycosylation mutants.

Paragraph starting: *'All investigated archaeal S-layer proteins are heavily glycosylated and therefore many of the essential functions of this surface protein might be mediated by their glycans...'*

8) No mention of S-layer removal being a first step in numerous strategies used to transform a variety of archaeal strains.

We agree that this aspect is also worth mentioning and have added a sentence: *'As a first barrier, the S-layer potentially shields the cell from the environment, as exemplified by standard transformation protocols for halophilic and methanogenic Archaea, which require transient removal of the S-layer in order to facilitate DNA uptake'* (Leigh et al., 2008).

9) Line 81: Misleading. A role for S-layer glycosylation was reported in viral infection of a halophilic archaea by Eichler and Roine (ref 63 in this manuscript).

We have extensively commented on these studies in an extra paragraph, starting: *"Also, recent studies revealed S-layer glycosylation to be crucial for cell-to-cell interactions during the mating process of the archaeon Haloferax ..."*

10) Line 188: Can the authors provide statistics on how many cells they observed lacking gaps in the S-layer and the sizes of these gaps? While the figures shown are impressive, they are single examples.

We agree and have re-examined previously captured as well as new electron micrographs to provided statistics. We have found cells with gaps or detached S-layer in approx. 8% of the thin sectioned cells (approx. n= 70). These gaps were never observed in the control cultures. This percentage might seem low at first glance, but considering that thin sections only allow for the visualization of a tiny fraction of the actual cell surface, the number of affected cells in the population must be far higher than what can be visualized with this method.

We have also measured gap sizes and detached S-layer lattices in thin sectioned samples. Gap widths were of varying sizes with an average of 359µm (+/-271µm,

(n=55). The results are included in the manuscript and shown in the Supplementary (new Supplementary Fig. 11).

11) Line 199: Has S-layer morphogenesis been studied in Archaea?

No. S-layer morphogenesis has only been studied in the Bacteria so far. We clarified that in the respective section.

12) General comment: Why not assign the different strains names that mean something to the reader? MoE-CC – surely, there is a friendlier way to name this strain.

We welcome the suggestion. The name of the control cultures/constructs and respective complementation constructs were changed to Ctrl (Control) and Ctrl-CC, respectively.

13) Line 202: It is not clear what the authors are seeing. There is no control for the specificity of lectin (not identified here, only in the fig legend) binding (e.g. upon competition with added sugar, after chemical deglycosylation). Has it been shown that this particular ligand binds the glycan attached to *S. solfataricus* proteins? Glycolipids? The equivalent of a western blot using the lectins would be welcome. Does the decrease in glycosylation correspond with the extent of SlaB deletion? Why would introduced *S. islandicus* SlaB be differently glycosylated in *S. solfataricus* than would the native SlaB? Host-dependent glycosylation in Archaea has been demonstrated.

We thank the reviewer for these constructive remarks.

We employed WGA, because it binds to N-acetylglucosamine, which in previous studies has been shown to be one of the building blocks of *S. solfataricus* S-layer glycan structure (Palmeri *et al.* 2013). To verify that WGA-Alexa488 conjugate binds indeed to the S-layer glycan of *S. solfataricus* we have performed an additional experiment.

We produced ghost cells (lysed cells composed only of S-layer) according to Veith *et al* 2009. A sample of an exponentially growing Ctrl culture (OD₆₀₀=0.1) and a sample of ghost cells were incubated with WGA-Alexa488 as well as DAPI and Nile Red to test for the presence of N-acetylglucosamine, DNA and lipids, respectively. As can be seen in new Supplementary Fig. 6, the WGA-Alexa488 stain shows similar fluorescence in both samples, whereas DAPI and Nile Red which produce a clear fluorescence in the Ctrl sample, show no or almost no fluorescence in the ghost cells. This shows that the Lectin stain does indeed bind to the N-acetylglucosamine in the glycans decorating the *S. solfataricus* S-layer proteins. In order to address the possibility of competition by sugars in the supernatant, we repeated the whole experimental series (new Fig. 4) with freshly grown cells with the following modification: Cells were diluted 1:20 in Tris-Acetate pH 5.5 before the WGA-Alexa488 was added and the lectin was added in excess. We changed the protocol accordingly in Materials and Methods.

14) Does the decrease in glycosylation correspond with the extent of SlaB deletion?

We thank for this comment and agree that a gradual decrease in glycosylation paralleling extent of SlaB deletion is more convincing. We have therefore repeated the glycosylation experiments with samples from a new set of transformed cultures that included transformants with different levels of depletion. Indeed, the decrease in glycosylation corresponded to the extent of SlaB deletion.

We have updated the results in the main manuscript (new Fig. 4 and new Supplementary Fig. 8).

15) Line 211: The glycosylated nature of SlaB was previously reported (ref 51 in this manuscript).

We apologize for this misunderstanding. We were addressing the different quantities of glycosylation sites (sequons) on the proteins and not a difference in glycan structure. The predicted differences in number of sequons between *Sulfolobus* species has been addressed by Meyer *et al.* 2013 and Palmeri *et al.* 2013. We therefore believe that a different quantity of glycosylation sites could explain the shift in fluorescence intensity.

We have made this more clear in the text now.

16) General question: Despite a reduced level of SlaB, is there no compensating self-assembly of SlaA? The authors mention the possibility of non-membrane-anchored SlaA-based constructs in the medium later in the Discussion but provide no evidence.

We thank the reviewer for this interesting question. Recent preliminary data show that the reassembly of SlaA in solution is indeed possible. Interestingly, the detached sheaths of assembled SlaA that we observed formed without the presence of SlaB by self-assembly. However, we consider it more likely that SlaA assembled into the native lattice with the aid of the remaining, membrane bound SlaB. Since these results are still under investigation, we only make a statement in the discussion, but not in the result section. Additional data have also been added to the Supplementary Data. (see new Supplementary Methods 1).

17) General comment: Albers has shown that in *S. acidocaldarius*, protein glycosylation is essential for survival. It was proposed that the water-trapping sugar coating on the cell surface offers an advantage in a thermoacidic environment. Could the effects seen in this manuscript not be due to decreased surface glycosylation resulting from the perturbed S-layer rather than the perturbed S-layer itself? This could be tested by using the CRISPR technology employed here on enzymes of glycosylation (e.g. AgIB).

We agree that it would be very interesting to dissect which substructure of the S-layer is responsible for the strong phenotypes that we observe. However, silencing the expression of AgIB or other glycosylating components might also not give a clear answer, since other outer surface structures are also glycosylated by the same components, so one cannot exclude pleiotropic effects. Only more complex investigations, perhaps involving mutations of proteins combined with gene silencing experiments could perhaps resolve this question, but we consider this to be beyond the scope of this paper

.18) Line 234: Aren't these finding opposite of those reported by Albers (Meyer et al 2013)?

No, we don't think that our data contradict. They represent different experiments. Meyer *et al.*, in 2013 found that 16Agl – glycosyltransferase knockout strains suffered from severe growth retardation under elevated salt concentrations (200 – 400mM). They measured growth by analysing the optical density of a culture which does not discriminate between increase in cell number and cell size (i.e. diameter). However, in our study, morphological changes on the cell level upon hyperosmotic stress (mainly shrinking of cells) were investigated. As the goals of these studies were different, we do not believe that they are comparable as such. However, we do not dispute that higher

salt concentrations would potentially compromise growth (probably even more than seen in normal medium) of S-layer depleted cells, as we do observe a response to elevated salt concentrations in form of a decrease in the average cell size (even if only mildly).

19) General question: Are the SlaB-lacking cells motile? Assessing this would provide novel information on the importance of the S-layer for archeallum construction, stability and function. Such studies would help distinguish the importance of the S-layer as opposed to S-layer glycosylation.

We thank the reviewer for this comment and for providing the motivation to perform such experiments. We conducted a motility assay according to Meyer *et al*, 2013 for *S. acidocaldarius*. 10^7 cells were dotted on semi-solid gelrite plates containing basal brock only (without sugar, nitrogen source) and brock supplemented with 0,001% tryptone.

We observed small halos on plates without tryptone with SlaB depleted (4 mm) and control cells (5mm). However, as our results are not that clear and the assay might need further adaptation for *S. solfataricus*, we decided not to include this result in the paper.

20) Line 376: Any support that the strain actually lives in the bubbling hot spring and not where no bubbling? No clusters or biofilms in nature?

We agree that *Sulfolobus solfataricus* might also thrive in sediments or without bubbling, but all dozens of strains isolated by Zillig and collaborators (including one of us) were isolated from the inner, most strongly bubbling zone of hot springs (for example those described in Zillig *et al.* 1994, Systematic and Applied Microbiology, 16, 609-628) but also *Sulfolobus solfataricus* P1 used in this study. In contrast, *S. acidocaldarius* was probably isolated from the edge of hot springs (by Thomas Brock in the 1980s).

REVIEWERS' COMMENTS:

Reviewer #1 (Remarks to the Author):

Zink et al. addressed most of the comments through additional data or text modification. However, the key issue regarding the novelty of the work still remains. While the Whitaker group has successfully deleted the *slaA* and *slaB* genes in *S. islandicus* M.16.4, Zink et al tend to attribute this to genetic compensation reactions, which is wrong and misleading. As shown in a following work from the Whitaker group, which has been accepted by mBIO (<https://www.biorxiv.org/content/10.1101/444406v1>), the *slaA* and *slaB* knockout strains demonstrated very similar phenotype as the knockdown strains described here by Zink et al. (complete or partial lack of S-layer; increased cell size and DNA content in some cells etc). This indicates that the successful deletion of *slaA* and *slaB* was not owing to the so-called genetic rescue, but rather demonstrates the non-essentiality of the two genes.

Moreover, the virus infection data is not very convincing. First, the spotting assay presented in Figure 7a did not show significant difference between the control and the knockdown strain. Second, the large SD values (Figure 7b) from the extracellular viral DNA quantification make it difficult to judge the claimed difference between the control and the knockdown strain.

Reviewer #2 (Remarks to the Author):

All my concerns have been addressed.

REVIEWERS' COMMENTS:

Reviewer #1:

Zink et al. addressed most of the comments through additional data or text modification. However, the key issue regarding the novelty of the work still remains. While the Whitaker group has successfully deleted the *slaA* and *slaB* genes in *S. islandicus* M.16.4, Zink et al tend to attribute this to genetic compensation reactions, which is wrong and misleading. As shown in a following work from the Whitaker group, which has been accepted by mBIO (<https://www.biorxiv.org/content/10.1101/444406v1>), the *slaA* and *slaB* knockout strains demonstrated very similar phenotype as the knockdown strains described here by Zink et al. (complete or partial lack of S-layer; increased cell size and DNA content in some cells etc). This indicates that the successful deletion of *slaA* and *slaB* was not owing to the so-called genetic rescue, but rather demonstrates the non-essentiality of the two genes.

*The phenotypes observed by the above-mentioned studies were partly similar. However, there are striking differences, such as that we observe a strong size difference in the *slaB* depletion cultures, as well as a significant growth retardation (which fits well with the cell division defect observed in our study) and no cell aggregation. We have addressed the different outcomes of the studies in the discussion and provided possible explanations. We still believe that one reasonable explanation of the strongly differing phenotypical effects might be due to the different cultivation conditions (i.e. shaking in our case vs. non – shaking in the Zhang study). The observed cell aggregates in the Zhang experiment, which might form in order to protect the “naked” cells (and therefore potentially increases the survival rate), support such an explanation, as non-shaking conditions promote cell clustering. We therefore believe that the S-layer can be essential for cells under certain conditions, as e.g. in planktonic lifestyle.*

Moreover, the virus infection data is not very convincing. First, the spotting assay presented in Figure 7a did not show significant difference between the control and the knockdown strain. Second, the large SD values (Figure 7b) from the extracellular viral DNA quantification make it difficult to judge the claimed difference between the control and the knockdown strain.

Concerning the high SD values: those are often observed in virus experiments in particular with SSV1, that tends to form clusters. We would like to point out that these experiments were repeated with different MOI (multiplicity of infections, see table 1) and the outcome was significant. For us, the spotting assay is not ambiguous but supports well the quantitative studies.

Reviewer #2:

All my concerns have been addressed.